# The influence of biological, epidemiological, and treatment factors on the establishment and spread of drug-resistant *Plasmodium falciparum*

**Thiery Masserey[1,2], Tamsin Lee[1,2], Monica Golumbeanu[1,2], Andrew J Shattock[1,2], Sherrie L Kelly[1,2], Ian M Hastings[3], Melissa A Penny[1,2]***

[1]Swiss Tropical and Public Health Institute, Allschwil, Switzerland; [2]University of Basel, Basel, Switzerland; [3]Liverpool School of Tropical Medicine, Liverpool, United Kingdom

**Abstract** The effectiveness of artemisinin-based combination therapies (ACTs) to treat *Plasmodium falciparum* malaria is threatened by resistance. The complex interplay between sources of selective pressure—treatment properties, biological factors, transmission intensity, and access to treatment—obscures understanding how, when, and why resistance establishes and spreads across different locations. We developed a disease modelling approach with emulator-based global sensitivity analysis to systematically quantify which of these factors drive establishment and spread of drug resistance. Drug resistance was more likely to evolve in low transmission settings due to the lower levels of (i) immunity and (ii) within-host competition between genotypes. Spread of parasites resistant to artemisinin partner drugs depended on the period of low drug concentration (known as the selection window). Spread of partial artemisinin resistance was slowed with prolonged parasite exposure to artemisinin derivatives and accelerated when the parasite was also resistant to the partner drug. Thus, to slow the spread of partial artemisinin resistance, molecular surveillance should be supported to detect resistance to partner drugs and to change ACTs accordingly. Furthermore, implementing more sustainable artemisinin-based therapies will require extending parasite exposure to artemisinin derivatives, and mitigating the selection windows of partner drugs, which could be achieved by including an additional long-acting drug.

**\*For correspondence:** melissa.penny@unibas.ch

**Competing interest:** The authors declare that no competing interests exist.

## Editor's evaluation

The authors provide an analysis of how various factors (biological, epidemiological, and treatment) impact the establishment and spread of drug-resistant *Plasmodium falciparum* using a combination of transmission modeling and model emulation. This comprehensive approach to investigating the complex dynamics underlying drug resistance explicitly considers several factors, highlighting their roles in the increasingly important public health question relating to spread of drug-resistant *Plasmodium falciparum*.

## Introduction

Malaria remains a global health priority (**WHO, 2020a**). The WHO recommends several artemisinin-based combination therapies (ACTs) to treat uncomplicated *Plasmodium falciparum* malaria (**WHO, 2020b**). ACTs combine a short-acting artemisinin derivative to rapidly reduce parasitaemia during the first 3 days of treatment and a long-acting partner drug to eliminate remaining parasites (**WHO,**

*2020b*). These drug combinations are intended to delay the evolution of drug resistance, which has frequently occurred under monotherapy treatment (*Farooq and Mahajan, 2004*; *White, 1999*; *White, 2004*; *Wongsrichanalai et al., 2002*). However, parasites partially resistant to artemisinin have emerged in the Greater Mekong Subregion (GMS) and, more recently, in Rwanda, Uganda, Guyana, and Papua New Guinea despite the use of ACTs (*WHO, 2020b*; *Chenet et al., 2016*; *Uwimana et al., 2020*; *Uwimana et al., 2021*; *Miotto et al., 2020*; *Balikagala et al., 2021*). Partial artemisinin resistance leads to slower parasite clearance following treatment with ACTs, but not necessarily to treatment failure (*WHO, 2020b*). However, high rates of treatment failure have been observed in the GMS due to parasites being less sensitive to artemisinin derivatives and their partner drugs (*WHO, 2020b*). To prevent the evolution of drug-resistant parasites and to preserve the efficacy of ACTs or triple artemisinin-based combination therapies (TACT, including a second long-acting drug) now being tested (*van der Pluijm et al., 2020*), it is essential to understand which factors drive this process.

The evolution of drug resistance follows a three-step process of mutation, establishment, and spread. First, mutations conferring drug resistance emerge in the population at a rate that depends on multiple factors, such as organism mutation and migration rates (*zur Wiesch et al., 2011*; *Mackinnon, 2005*). Second, establishment is a highly stochastic step as the parasite with the drug-resistant mutation needs to infect other hosts (*zur Wiesch et al., 2011*; *Mackinnon, 2005*; *Hastings, 2004*; *Hastings et al., 2020*). The resistant strain establishes in the population once its frequency is high enough to minimise its risk of stochastic extinction (*zur Wiesch et al., 2011*; *Mackinnon, 2005*; *Hastings, 2004*; *Hastings et al., 2020*). Several forces influence the establishment of mutations. In settings with higher heterogeneity of parasite reproductive success, establishment of mutations is less likely because the effects of stochasticity are more substantial (*zur Wiesch et al., 2011*; *Hastings, 2004*; *Hastings et al., 2020*; *Hastings and Mackinnon, 1998*). This heterogeneity depends on the level of transmission and health system strength (*zur Wiesch et al., 2011*; *Hastings, 2004*; *Hastings et al., 2020*; *Hastings and Mackinnon, 1998*; *Klein, 2014*). In addition, the more selection favours the resistant strain, the more likely it is to establish (*zur Wiesch et al., 2011*; *Hastings, 2004*; *Hastings et al., 2020*; *Hastings and Mackinnon, 1998*). The strength of selection depends on many factors, such as the parasite and human biology, the transmission setting, drug properties, and health system strength (*White, 2004*; *Antao and Hastings, 2011*; *Hughes and Andersson, 2015*; *Huijben et al., 2013*; *Miotto et al., 2015*; *Slater et al., 2017*; *Mackinnon and Marsh, 2010*). Third, resistance spreads through a region after a resistant mutation has become established. The mutation spreads at a rate that depends on the strength of selection (*zur Wiesch et al., 2011*; *Hastings et al., 2020*).

It is not fully understood how factors intrinsic to the transmission setting, health system, human and parasite biology, and drug properties interact to influence the establishment and spread of drug-resistant parasites. Mathematical models of infectious disease have not previously been used to systematically assess the joint influence of multiple factors on the establishment and spread of drug-resistant *P. falciparum* (e.g. *Klein, 2014*; *Slater et al., 2017*; *Bushman et al., 2018*; *Whitlock et al., 2021*; *Hastings et al., 2002*; *Watson et al., 2021*; *Brock et al., 2018*; *Watkins and Mosobo, 1993*; *Pongtavornpinyo et al., 2008*; *White et al., 2009*; *Chiyaka et al., 2009*; *Esteva et al., 2009*; *Koella and Antia, 2003*; *Lee and Penny, 2019*; *Lee et al., 2022*; *Legros and Bonhoeffer, 2016*; *Tchuenche et al., 2011*; *Tumwiine et al., 2007*) (and to the best of our knowledge other drug-resistant pathogens [virus, bacteria, etc.]). Simple models, based on the Ross and MacDonald model (*Macdonald, 1957*; *Ross, 1915*), have considered specific components of the epidemiology of resistance and, therefore, are not sophisticated enough to answer questions on how factors have jointly impacted establishment and spread of drug resistance (*Brock et al., 2018*; *Chiyaka et al., 2009*; *Esteva et al., 2009*; *Koella and Antia, 2003*; *Lee and Penny, 2019*; *Tchuenche et al., 2011*; *Tumwiine et al., 2007*). Most models have investigated specific transmission scenarios and questions, such as how within-host competition between parasites influences development of drug resistance (*Bushman et al., 2018*; *Pongtavornpinyo et al., 2008*; *Legros and Bonhoeffer, 2016*), and did not systematically assess the impact of assumptions used on their results. Consequently, previous studies have neither systematically compared the influence of multiple drivers, nor assessed how their influence varies under different transmission settings or health system strengths.

In addition, most models have made simplifications concerning drug action and consequences of partial resistance. They have typically not explicitly modelled the pharmacokinetics and pharmacodynamics of the drugs and have assumed that resistant parasites are fully resistant to the drugs. Parasites

partially resistant to artemisinin exhibit an extended ring-stage during which they are not sensitive to artemisinin; however, parasites remain sensitive to artemisinin during other stages (*Klonis et al., 2013*; *Wang et al., 2017*; *Sá, 2018*; *Witkowski et al., 2013*; *Ye et al., 2016*). In addition, parasites resistant to partner drugs have an increased minimum inhibitory concentration (MIC), meaning that they are not sensitive to low drug concentrations but remain susceptible to high concentrations of partner drugs (*Chaorattanakawee et al., 2016*; *Chaorattanakawee et al., 2015*; *Tahita et al., 2015*). Consequently, many models have ignored the residual effect of drugs on resistant parasites and have not investigated the influence of the degree of resistance and drug proprieties on the establishment and spread of drug resistance. Models that have explicitly considered drug action have focused on specific questions such as how half-life impacts the spread of resistance or how resistance to the partner drug influences evolution of artemisinin resistance (*Hastings et al., 2002*; *Watson et al., 2021*). However, they did not investigate how the impact of drug proprieties and the degree of resistance interact with other biological, transmission, and health system factors.

In this study, we developed a disease model with an emulator-based approach to quantify the influence of factors intrinsic to the biology of the parasite and human, the transmission setting, the health system strength, and the drug properties on the establishment and spread of drug-resistant parasites. Our approach is based on a detailed individual-based malaria model, OpenMalaria (https://github.com/SwissTPH/openmalaria/wiki), that includes a mechanistic within-host model (based on *Molineaux et al., 2002*). We first adapted our model, OpenMalaria, to explicitly include mechanistic drug action models at the individual level (as a one, two, or three-compartment pharmacokinetic model with a pharmacodynamics component of parasite killing [*Bertrand and Mentré, 2008*; *Kay et al., 2013*; *Johnston et al., 2014*; *Winter and Hastings, 2011*]) and to track multiple parasite genotypes to which we could assign fitness costs and drug susceptibility (i.e. pharmacodynamics) properties. We then built an emulator-based workflow to quantify, through a series of global sensitivity analyses, the influence of multiple factors on the establishment and spread of parasites having different degrees of resistance to artemisinin derivatives and/or their partner drugs when used in monotherapy and combination (as ACTs). Emulators are predictive models that can approximate the relationship between input and output parameters of complex models and can run much faster than complex models to perform global sensitivity analyses more efficiently (*Grow and Hilton, 2018*). OpenMalaria is a mechanistic model, so the observed dynamics at the population level (e.g. the spread of resistant genotypes) emerges from the relationship between the different model components and their input parameters. These dynamics can only be understood and tested through extensive analyses as undertaken here. Identifying which factors (e.g. drug properties and/or setting characteristics) favour the evolution of resistance, enables us to identify drug properties or strategies to slow or mitigate resistance and guides the development and implementation of more sustainable therapies.

## Results

### Development of drug resistance

We investigated the establishment and spread of drug-resistant genotypes by varying the degrees of resistance for three different treatment profiles. The first treatment profile considered was a monotherapy using a short-acting drug. The short-acting drug had a short half-life and a high killing efficacy typical of artemisinin derivatives (*Figure 1A and B*). Patients received a daily dose of the short-acting drug for 6 days (see Materials and methods). To mimic the mechanism of resistance to artemisinin derivatives, we assumed that genotypes resistant to the short-acting drug had lower maximum killing rates (Emax) than sensitive ones (*Figure 1B*) (see Materials and methods). We defined the degree of resistance to the short-acting drug as the relative decrease of the Emax of the resistant genotype compared with the sensitive one. The second treatment profile was also a monotherapy but with a long-acting drug. The long-acting drug had a longer half-life and a lower Emax than the short-acting drug, typical of partner drugs used for ACTs (such as mefloquine, piperaquine, and lumefantrine) (*Figure 1A and B*). Patients received a daily dose of the long-acting drug for 3 days (see Materials and methods). We assumed that genotypes resistant to the long-acting drug had higher half-maximal effective concentrations (EC50) than sensitive ones (*Figure 1B*) (see Materials and methods). We defined the degree of resistance to the long-acting drug as the relative increase of the EC50 of the resistant genotype compared with the sensitive genotype. Note that monotherapies for malaria are

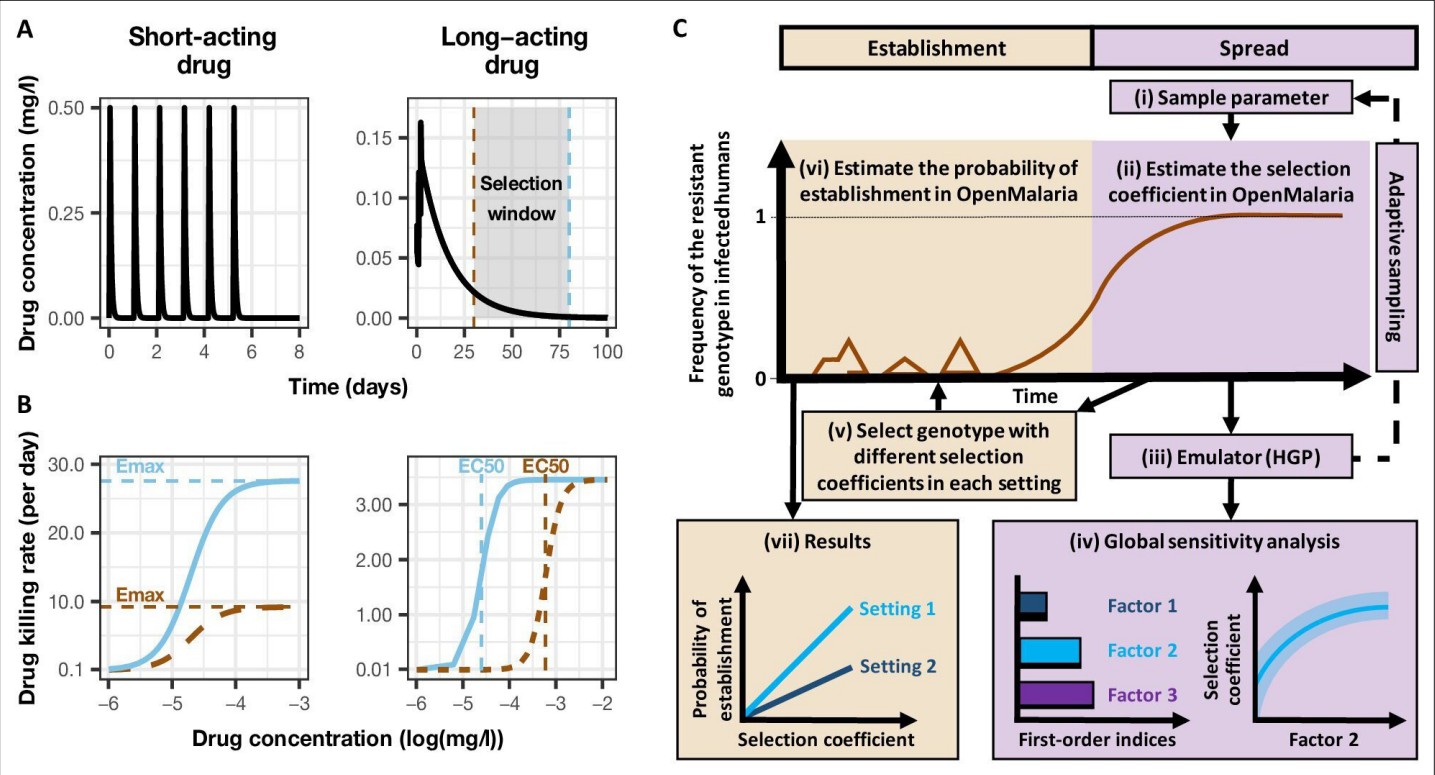

**Figure 1.** Overview of treatment profiles and the modelling workflow. (**A**) Examples of the modelled within-host concentration (mg/l) of both the short- and long-acting drugs used as monotherapy. Here, patients received a daily dose of the short-acting drug for 6 days or a daily dose of the long-acting drug for 3 days. The grey shaded area represents an exemplar selection window (defined as the period of time post-treatment when drug concentration is sufficiently high to prevent reinfection by drug-sensitive infections but is sufficiently low to allow reinfection by drug-resistant infections). The short- and long-acting drugs used in combination (like ACTs) had the same respective profile as in monotherapy, but patients received a daily dosage of each drug over 3 days, as recommended by WHO for ACTs (**WHO, 2021**). (**B**) Illustrations of the modelled relationship between the concentration (log[mg/l]) and the killing effect (per day) of the short- and long-acting drugs on the resistant (brown dashed curve) and sensitive genotypes (solid blue curve). Compared with sensitive genotypes, resistant parasites had a reduced maximum killing rate (Emax) when resistant to the short-acting drug and an increased half-maximal effective concentration (EC50) when resistant to the long-acting drug. (**C**) Schematic of the modelling workflow: central plot, brown curve represents an exemplar frequency of the resistant genotype in infected humans. The purple area (right side) shows the steps for assessing the influence of factors on the rate of spread (selection coefficient) of a resistant genotype through global sensitivity analysis of an emulator trained on our model simulations (see Materials and methods). In brief: (i) randomly sampling combinations of parameters, (ii) assessing the rate of spread of the resistant genotype for each parameter combination, (iii) training an emulator to learn the relationship between the input (for the different drivers) and output (the rate of spread) with iterative improvements to fitting through adaptive sampling, (iv) performing the global sensitivity using the trained emulator. The global sensitivity analysis estimates both first-order indices of each factor (representing their influence on the rate of spread) and the 25th, 50th, and 75th quantiles of the estimated selection coefficient from the emulator across each parameter range. The orange area (left side) shows the steps to assess the relationship between the selection coefficient and the probability of establishment in different transmission settings (see Materials and methods). In brief: (v) selecting genotypes with different selection coefficients in each setting, (vi) assessing their probability of establishment, and (vii) visualising the relationship between the probability of establishment and the section coefficient in each setting. HGP: Heteroskedastic Gaussian Process.

no longer recommended, but we investigated drivers of resistance under monotherapy to identify the determinants specific to each drug profile. The last treatment profile was a daily dose of a combination of the short-acting and the long-acting drugs for 3 days, simulating ACTs. In this case, we focused on resistance to the short-acting drug, as artemisinin is the shared compound of all ACTs and is of greater concern. Thus, the resistant and sensitive genotypes refer to sensitivity to the short-acting drug. We measured selection for resistance to the short-acting drug against a background of differing sensitivity to the long-acting partner drug, whose effectiveness was varied as described in *Table 1*. We assumed that the genotypes sensitive and resistant to the short-acting drug had identical sensitivities to the partner drug (i.e. there was no cross-resistance).

Our analysis had two steps. First, we quantified the impact of factors listed in *Table 1* on the spread of drug-resistant parasites through global sensitivity analyses using an emulator trained on our

**Table 1.** Potential drivers of the spread of drug resistance.

List of factors and their parameter ranges investigated in the global sensitivity analyses of the spread of parasites resistant to each treatment profile. The parameter ranges were defined based on the literature as described in Materials and methods. Note that the parameter ranges of the short-acting drug captured the parameter values of typical artemisinin derivatives, and the parameter ranges of the long-acting drug captured the parameter values of partner drugs of artemisinin derivatives such as mefloquine, piperaquine, and lumefantrine. In addition, note that the ratio maximum drug concentration/half-maximal effective concentration (Cmax/EC50) is not a direct input of the model, but we varied this ratio by varying the EC50 of the sensitive genotype and the drug dosage (which impacted Cmax). We initially assessed the effect of Cmax and EC50 on the rate of spread independently; however, we found that the impact of the EC50 and the Cmax on the drug killing effect post-treatment depended on their ratio (see Materials and methods). A Latin hypercube sampling (LHS) algorithm was used to sample from the ranges of all parameters (*Gramacy, 2007*).

| Component | Determinant | Definition | Parameter range (References) | |
|---|---|---|---|---|
| | | | Short-acting drug | Long-acting drug |
| Drug properties (PK/PD model) | Half-life | Time for the drug concentration to fall by 50% (days) | (0.035, 0.175) (*Kay et al., 2013*; *Winter and Hastings, 2011*) | (6, 22) (*Charles et al., 2007*; *Staehli Hodel et al., 2013*; *Jullien et al., 2014*; *Karunajeewa et al., 2008*; *Maganda et al., 2015*) |
| | Emax | Maximum killing rate the drug can achieve (per day) | (27.5, 31.0) (*Kay et al., 2013*) | (3.45, 5.00) (*Winter and Hastings, 2011*) |
| | Cmax/EC50 | The ratio between the maximum drug concentration (Cmax) and the half-maximal effective concentrations (EC50) of the sensitive genotype. This calculated ratio captures the duration of the drug killing effect by capturing how high the Cmax is compared to the EC50 | (55.0–312.0) (*Kay et al., 2013*; *Winter and Hastings, 2011*) | (5.1–21.7) (*Kay et al., 2013*; *Winter and Hastings, 2011*) |
| Parasite biology | Degree of resistance (PK/PD model) | For the short-acting drug: relative decrease of the Emax of the resistant genotype compared with the sensitive one. For the long-acting drug: relative increase of the EC50 of the resistant genotype compared with the sensitive one (see Materials and methods) | (1, 50) | (1, 20) |
| | Fitness cost | Relative reduction of the resistant genotype multiplication rate within the human host compared to the sensitive one | (1.0, 1.1) (*Kublin et al., 2003*; *Mita et al., 2003*) | |
| Transmission level | Entomological inoculation rate | Mean number of infective mosquito bites received by an individual during a year (inoculations per person per year) | (5, 500) (*Edwards et al., 2019a*; *Hay et al., 2000*; *Chaumeau et al., 2018*; *Edwards et al., 2019b*; *Yamba et al., 2020*) | |
| Health system | Level of access to treatment | The probability of symptomatic cases to receive treatment within two weeks from the onset of symptom onset (%) | (10, 80) | |
| | Diagnostic detection limit | Parasite density for which the probability of having a positive diagnostic test is 50% (parasites/μl) | (2, 50) (*Kilian et al., 2000*; *Murray et al., 2008*) | |

PK/PD: pharmacokinetic-pharmacodynamics; Cmax: maximum drug concentration; EC50: half-maximal effective concentration.

model simulations (*Figure 1C*, purple area [right side], see Materials and methods). For each simulation, we tracked a drug-sensitive genotype and a drug-resistant genotype, and we estimated the rate of spread using the selection coefficient, which measures the rate at which the logit of the resistant genotype frequency increases each parasite generation (see Materials and methods, note that a selection coefficient below zero implies that resistance does not spread in the population) (*Hastings et al., 2020*). Then, we assessed the probability of establishment for a subset of resistant genotypes with known and positive selection coefficients to observe the relationship between selection coefficient and the probability of establishment in different settings (*Figure 1C*, orange area [left side], see Materials and methods). We could then extrapolate the probability of establishing any mutations with a known selection coefficient, which made the process more efficient since estimating the probability of establishment requires running many more stochastic realisations than estimating the selection coefficient due to the stochasticity of this step.

## Key drivers of the spread of drug-resistant parasites

Under monotherapy, access to treatment (the probability of symptomatic cases to receive treatment within 2 weeks from the onset of symptoms) and degree of resistance of a monotherapy were the main drivers of the spread of resistance (*Figure 2A*). For the short-acting and the long-acting drugs used as monotherapy, the selection coefficient increased with increasing access to treatment (*Figure 2—figure supplement 1*). In addition, higher degrees of resistance of the resistant genotype to the short-acting drug (relative decrease in the resistant genotype Emax compared with the sensitive one) and the long-acting drug (relative increase in the resistant genotype EC50 compared with the sensitive one) promoted the spread of parasites resistant to the short-acting and the long-acting drugs, respectively (*Figure 2—figure supplement 1*).

When the short-acting and the long-acting drugs were used in combination in our model, we referred to the resistant and sensitive genotypes as the genotypes resistant and sensitive to the short-acting drug, respectively. However, both genotypes could have some degree of resistance to the long-acting drug. In this case, the most important driver of spread was the degree of resistance of both genotypes to the long-acting drug (*Figure 2A*). The median selection coefficient was below zero when both genotypes were susceptible to the long-acting drug (the minimum degree of resistance to the long-acting drug) (*Figure 2B*), indicating that using an efficient partner drug can limit the spread of artemisinin resistance. The spread of parasites resistant to the short-acting drug was accelerated when parasites were also resistant to the long-acting drug, highlighting that resistance to the long-acting drug can facilitate the spread of artemisinin resistance. We further illustrated with concrete examples (Appendix: section 1.1) how the spread of partial resistance to the short-acting drug accelerates with higher degrees of resistance to the long-acting drug. These results further confirmed that resistance to partner drugs facilitates the spread of resistance to artemisinin, highlighting the importance of combining artemisinin derivatives with an efficient partner drug.

## Variation in the influence of factors across settings and degrees of resistance

We compared the effects of drug properties and levels of fitness cost on estimated selection coefficients for a fixed set of degrees of resistance, levels of access to treatment, transmission intensities, seasonality patterns, and levels of adherence to treatment (percentage of treatment doses adhered by patients). *Figure 3* summarises the impact of key factors influencing estimated selection coefficients in seasonal transmission settings with a population fully adherent to treatment (the impact of factors was similar across seasonality pattern and levels of adherence to treatment *Figure 3—figure supplements 1–2*). The impact of all factors in each setting is shown in *Figure 3—figure supplements 1–2*.

Across settings with a low access to treatment, we found that fitness cost had the largest influence on the selection coefficient (*Figure 3—figure supplements 3–5*). The fitness cost of a resistant genotype was defined as the relative decrease in the resistant genotype multiplication rate within an untreated human host compared with the sensitive genotype. Consequently, high fitness costs prevented the spread of resistance (*Figure 3*, *Figure 3—figure supplement 2*). At a high level of access to treatment, the effect of fitness cost was reduced, and drug properties played a critical role in the spread of drug resistance, and their influence varied for each treatment profile as described below.

For the short-acting drug used as monotherapy, the half-life had an important influence on the rate of spread (*Figure 3—figure supplement 3*). A long half-life reduced the spread of resistant parasites by extending the period during which the drug killed partially resistant parasites (*Figure 3*, *Figure 3—figure supplement 1*). The half-life effect was stronger for parasites with a lower degree of resistance (*Figure 3*, *Figure 3—figure supplement 3*), as highly resistant parasites were less affected by the drug. Furthermore, the spread of the resistant genotype was faster in populations with low adherence to treatment (*Figure 3—figure supplement 6*) because with fewer treatment doses, the parasite was exposed to the drug for a shorter time, leading to higher parasite survival. Overall, these results highlight that the time during which the parasite is exposed to artemisinin is a critical driver of the spread of partial artemisinin resistance.

For parasites with a low degree of resistance to the long-acting drug used as monotherapy, the drug half-life also had a key influence on the selection coefficient (*Figure 3—figure supplement 4*). However, long half-lives were associated with large selection coefficients (*Figure 3*, *Figure 3—figure*

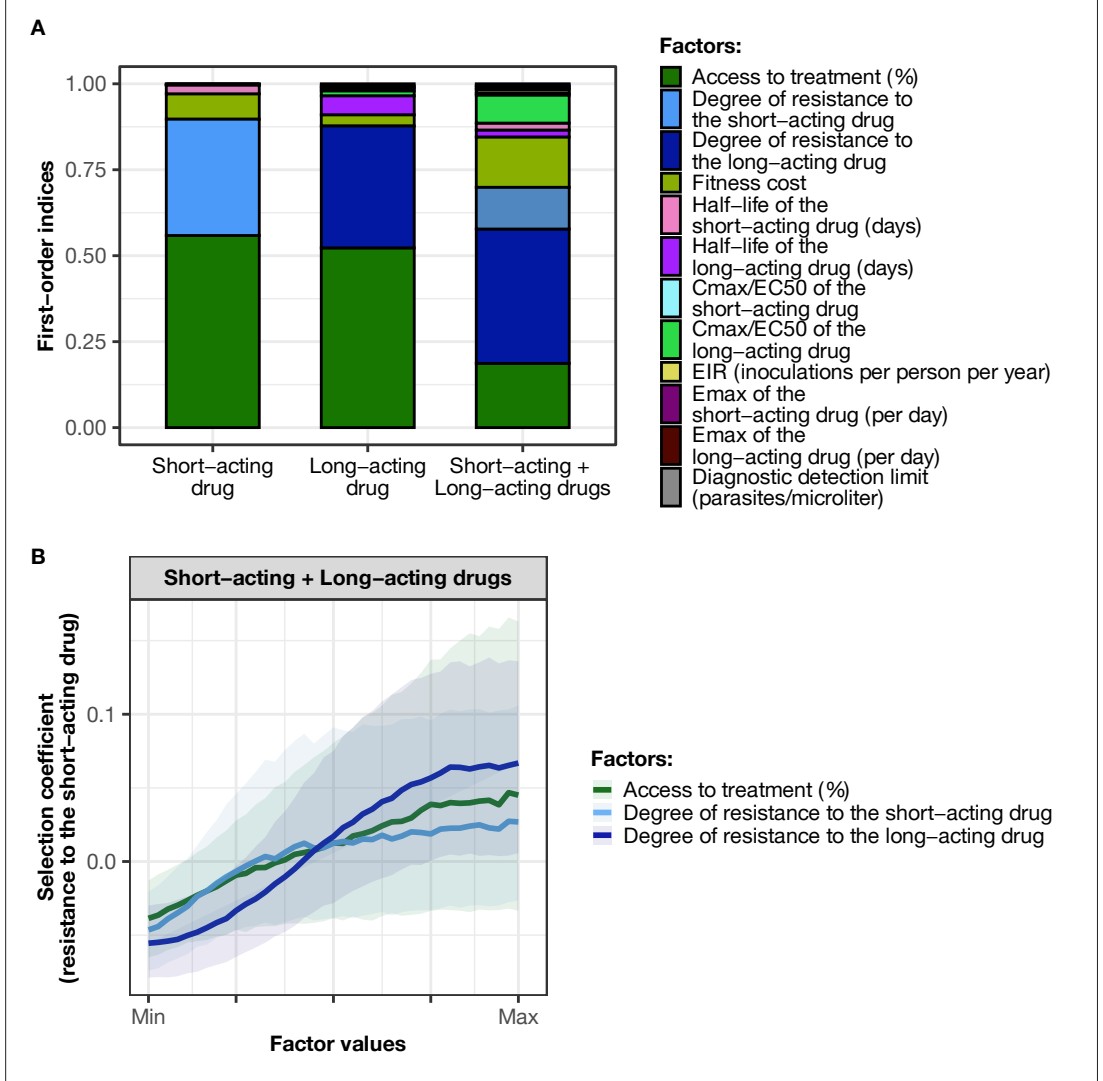

**Figure 2.** Influence of drug properties, fitness costs, degrees of resistance, transmission levels, and health system factors on estimated selection coefficients for three treatment profiles. (**A**) The first-order indices from our variance decomposition analysis indicate the level of importance of drug properties, fitness costs, degrees of resistance, transmission levels, access to treatment, and diagnostic limits in determining the spread of drug resistance. Indices are shown for each treatment profile in a non-seasonal setting with a population fully adherent to treatment. Selection coefficients are considered for the short-acting drug and the long-acting drug when each drug is used as monotherapy and for the short-acting drug when both drugs are used in combination. Definitions and ranges of parameters investigated are listed in *Table 1*. (**B**) Influence of factors on the selection coefficient of genotypes resistant to the short-acting drug in a population that used the short-acting and the long-acting drugs in combination. Curves and shaded areas represent the median and interquartile range of selection coefficients estimated during the global sensitivity analyses over the following parameter ranges: access to treatment (10–80%); the degree of resistance of the resistant genotype to the short-acting drug (1–50-fold reduction in Emax); and the degree of resistance of both sensitive and resistant genotypes to the long-acting drug (1–20-fold increase in EC50). A selection coefficient below zero implies that resistance does not spread in the population but is being lost due to its fitness costs. The transmission setting was non-seasonal and all treated individuals were fully adherent to treatment.

The online version of this article includes the following source data and figure supplement(s) for figure 2:

**Source data 1.** Related to *Figure 2A*.

**Source data 2.** Related to *Figure 2B*.

**Figure supplement 1.** Influence of the access to treatment and degree of resistance on the estimated selection coefficients of a genotype resistant to the short-acting drug or the long-acting drug used in monotherapy.

**Figure supplement 1—source data 1.** Related to *Figure 2—figure supplement 1*.

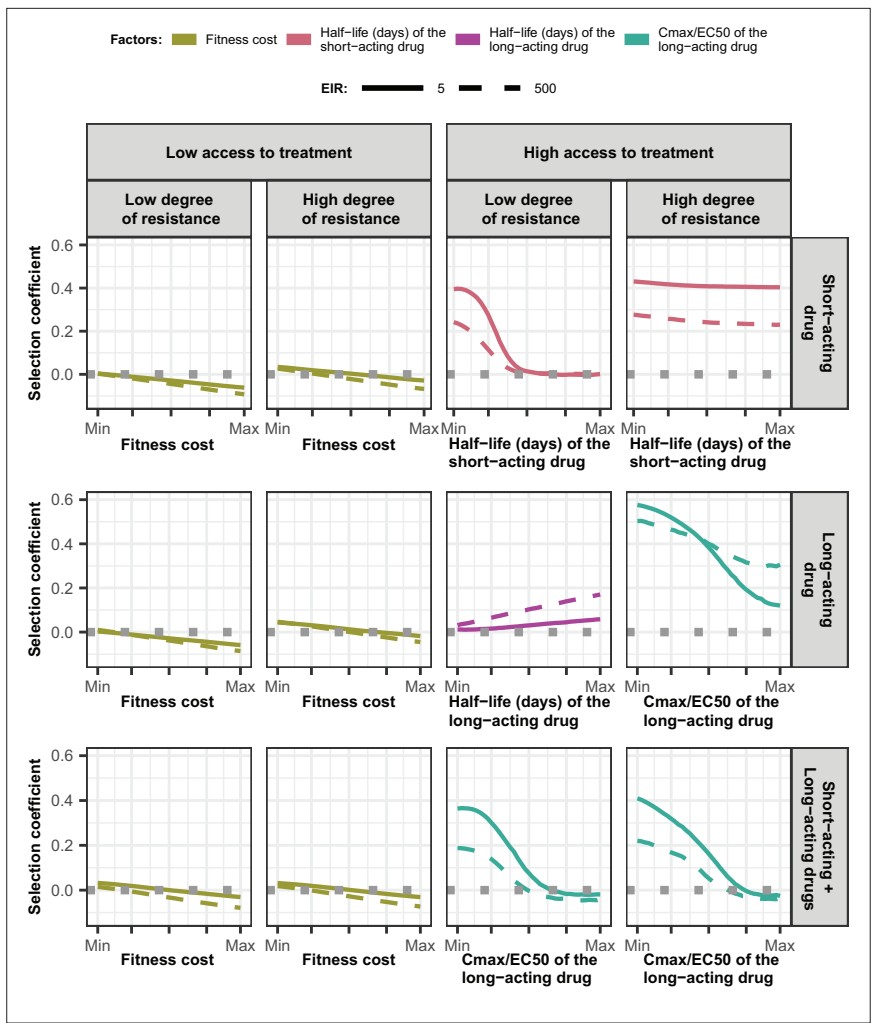

**Figure 3.** Magnitude and direction of effect of drug properties and fitness cost on estimated selection coefficients for low and high levels of transmission, degrees of drug resistance, and levels of access to treatment with monotherapy or combination treatment. The curves represent median selection coefficients over the parameter ranges of factors that were determined to have key influences on the rate of spread of drug-resistant genotypes in settings that had an entomological inoculation rate (EIR) of 5 (solid curves) or 500 (dashed curves) inoculations per person per year, and low (10%) or high (80%) levels of access to treatment. Selection coefficients illustrated the spread of parasites resistant to the short- and long-acting drugs when each drug was used as monotherapy and parasites resistant to the short-acting drug when both drugs were used in combination. For each treatment profile, results are shown for parasites with two different degrees of resistance; degree of resistance of 7 (low) and 18 (high) to the short-acting drug (Emax shift), 2.5 (low) and 10 (high) to the long-acting drug (EC50 shift), for the combination of the short-acting and the long-acting drugs, 7 (low) and 18 (high) to the short-acting drug and 10 to the long-acting drug. Results are illustrated for settings with a seasonality pattern of transmission and a population fully adherent to treatment. The impacts of all factors in all settings are shown in *Figure 3—figure supplements 1–2*. Parameter ranges are as follows: fitness cost (1.0–1.1); the half-life of the short-acting drug (0.035–0.175 days); the half-life of the long-acting drug (6–22 days); Cmax/EC50 ratio of the long-acting drug (5.1–21.7).

The online version of this article includes the following source data and figure supplement(s) for figure 3:

**Source data 1.** Related to *Figure 3*.

**Figure supplement 1.** Magnitude and direction of effect of drug properties and fitness cost on estimated selection coefficients in settings with high access to treatment and different levels of transmission, degrees of drug resistance, treatment adherence in seasonal, or perennial settings with monotherapy or combination treatment.

**Figure supplement 1—source data 1.** Related to *Figure 3—figure supplement 1*.

*Figure 3 continued on next page*

*Figure 3 continued*

**Figure supplement 2.** Magnitude and direction of effect of drug properties and fitness cost on estimated selection coefficients in settings with low access to treatment and different levels of transmission, degree of drug resistance, treatment adherence in seasonal, or perennial settings with monotherapy or combination treatment.

**Figure supplement 2—source data 1.** Related to *Figure 3—figure supplement 2*.

**Figure supplement 3.** First-order indices describing level of importance of each factor varied in the constrained sensitivity analysis of the spread of a genotype resistant to the short-acting drug used in monotherapy.

**Figure supplement 3—source data 1.** Related to *Figure 3—figure supplement 3*.

**Figure supplement 4.** First-order indices of each factor varied in the constrained sensitivity analysis of the spread of a genotype resistant to the long-acting drug used in monotherapy.

**Figure supplement 4—source data 1.** Related to *Figure 3—figure supplement 4*.

**Figure supplement 5.** First-order indices of each factor varied in the constrained sensitivity analysis of the spread of a genotype resistant to the short-acting drug when the short-acting drug and the long-acting drug are used in combination.

**Figure supplement 5—source data 1.** Related to *Figure 3—figure supplement 5*.

**Figure supplement 6.** Distribution of the estimated selection coefficient for resistant parasites with a low degree of resistance in different transmission settings with high access to treatment.

**Figure supplement 6—source data 1.** Related to *Figure 3—figure supplement 6*.

**Figure supplement 7.** Distribution of the estimated selection coefficients for resistant parasites with a high degree of resistance in different transmission settings with high access to treatment.

**Figure supplement 7—source data 1.** Related to *Figure 3—figure supplement 7*.

**Figure supplement 8.** Treatment usage.

**Figure supplement 8—source data 1.** Related to *Figure 3—figure supplement 8*.

**Figure supplement 9.** Distribution of selection coefficient of parasites with a low degree of resistance in different settings with low access to treatment.

**Figure supplement 9—source data 1.** Related to *Figure 3—figure supplement 9*.

*supplement 1*). Drugs with a long half-life have an extended period of low drug concentration in treated patients during which only resistant parasites can infect the host. This period of low drug concentration is called the selection window (*Hastings et al., 2002*; *Kay and Hastings, 2015*). These results confirm that the selection window plays a crucial role in the spread of resistance to long-acting drugs.

The spread of parasites with a high degree of resistance to the long-acting drug used as monotherapy was also accelerated by longer drug half-life (*Figure 3—figure supplement 4*). For these resistant parasites, the ratio Cmax/EC50 also had an important influence on the rate of spread (*Figure 3—figure supplement 4*). This ratio captured the duration of the drug killing effect on the sensitive genotype by assessing the proximity between the EC50 of the sensitive genotype and Cmax. A lower Cmax/EC50 ratio captures a shorter duration of the drug killing effect for the sensitive genotype and, consequently, also a lower duration of the drug killing effect against the resistant genotype (higher EC50). Thus, when the drug had a low Cmax/EC50 ratio, the duration of the drug killing effect was not long enough to ensure the successful clearances of parasites with a higher degree of resistance (higher EC50), favouring their spread (*Figure 3*, *Figure 3—figure supplement 1*). Furthermore, for parasites with a low degree of resistance (lower EC50), the ratio Cmax/EC50 also influenced the rate of spread in settings with a low level of treatment adherence, since low adherence reduces a Cmax leading to treatment failure (*Figure 3—figure supplements 1 and 4*). These results highlight the importance of treatment adherence to assure that the drug concentration is high enough to eliminate partially resistant genotypes and limit their spread.

When the genotype was resistant to the short-acting drug in a population that used the short-acting and the long-acting drugs in combination, factors related to the long-acting drug had the most influence on the selection coefficient (*Figure 3*, *Figure 3—figure supplement 5*). When the Cmax/EC50 ratio of the long-acting drug was large, the duration of the killing effect of the long-acting drug on parasites resistant to the short-acting drug was higher, reducing their spread (*Figure 3*, *Figure 3—figure supplement 1*). In addition, the rate of spread rose when the level of adherence to treatment

was low (*Figure 3—figure supplement 6*). These results highlight that the spread of partial resistance to artemisinin strongly depends on the capacity of the partner drug to kill them.

The influence of the transmission intensity (represented by entomological inoculation rate [EIR]) and its seasonality on the selection coefficient varied by treatment profiles and degrees of resistance. When the parasite was resistant to the short-acting drug (when used in monotherapy or combination), selection coefficients were higher in settings with lower EIR (*Figure 3—figure supplements 6 and 7*). We observed a similar trend for parasites with a high degree of resistance to the long-acting drug used in monotherapy (*Figure 3—figure supplement 7*). Two factors account for this trend. First, the selection of parasites resistant to the short-acting drug (low and high degrees of resistance) and parasites highly resistant to the long-acting drug depends on the proportion of treated infections. A higher portion of treated infections can lead to a higher proportion of delayed parasite clearance or treatment failure of drug-resistant infections allowing these resistant genotypes to spread. The proportion of treated infections is higher at lower EIR due to the lower level of immunity (*Figure 3—figure supplement 8*). Furthermore, lower immunity levels for individuals living in low transmission settings may also increase the risk of treatment failure and favour the spread of resistance. Second, there is a higher proportion of individuals coinfected by both genotypes at higher EIR. This higher level of co-infection enhanced within-host competition between genotypes, which inhibits the multiplication of resistant parasites within hosts due to their fitness cost and thus limits their spread. Similarly, the spread of resistant parasites was higher in the seasonal settings than in non-seasonal settings (*Figure 3—figure supplements 6 and 7*) due to the reduction of immunity levels and a decline in within-host competition between genotypes during the low transmission season of the seasonal settings. Overall, these results indicate that the spread of partial artemisinin resistance is faster in seasonal settings with low transmission levels.

However, for parasites with a low degree of resistance to the long-acting drug used in monotherapy, selection coefficients were higher in settings with a large EIR (*Figure 3—figure supplement 6*). This arises because the proportion of patients with low drug concentrations persisting from previous treatments increases at higher EIR where higher infection rates increase the overall usage of treatment (*Figure 3—figure supplement 8*). These low drug concentrations may fall within the selective window and hence drive the spread of parasites partially resistant to the long-acting drug. Note that this trend was only observed for settings with high access to treatment. In settings with

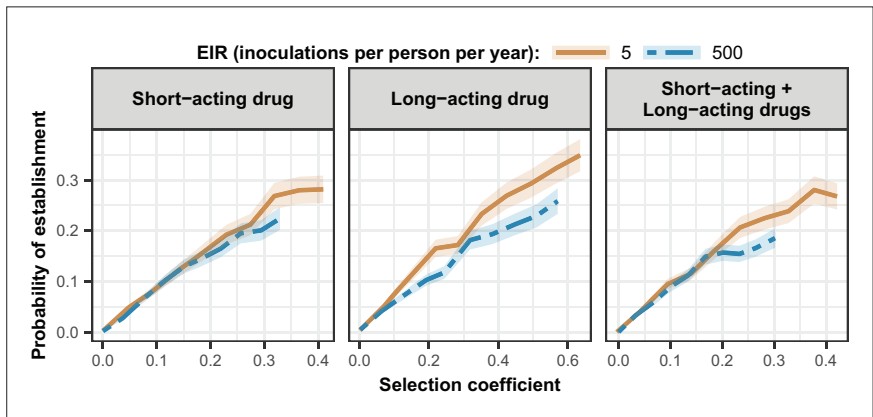

**Figure 4.** Estimated probability of establishment of mutations conferring drug resistance across transmission settings. Solid brown curves and blue dashed curves represent the relationship between the selection coefficient and the estimated probability of establishment of resistant parasites across settings that differ in transmission intensities (5 and 500 inoculations per person per year, respectively). The relationships are illustrated for parasites resistant to the short- and long-acting drugs when each drug was used as monotherapy and parasites resistant to the short-acting drug when both drugs were combined. The shaded area represents the 95% confidence intervals estimated as described in Materials and methods. The range of selection coefficients include higher values at a low entomological inoculation rate (EIR). For each setting, the level of access to treatment was specified as 80%, the population was assumed to be fully adherent to treatment (100%), and transmission was non-seasonal.

The online version of this article includes the following source data for figure 4:

**Source data 1.** Related to *Figure 4*.

low access to treatment, we observe similar trends as for parasites resistant to the short-acting drug (*Figure 3—figure supplement 9*) since here, the impact of the selection window was more negligible. These results highlight that the selection window of the long-acting drug can change the interplay between the transmission setting and the spread of drug resistance.

## Probability of establishment of drug resistance and its key drivers

Population genetic theory has shown that the probability of establishment of a mutation depends on two factors: (i) the size of its selection coefficient (i.e. establishment becomes more likely as the mutation becomes more advantageous) and (ii) the degree of heterogeneity in the number of parasite offspring. This occurs because higher heterogeneity increases stochastic fluctuations of allele number, so increases the chance that the mutation is lost despite its advantage (*zur Wiesch et al., 2011*; *Hastings, 2004*; *Hastings et al., 2020*; *Hastings and Mackinnon, 1998*). The probability of establishment can also be altered by temporal fluctuation in the population size or magnitude of the selection coefficient (*Waxman, 2011*). Both effects are likely to be present in seasonal settings of malaria transmission where population size fluctuates, and selection intensity may also change if the level of drug use fluctuates in response to the seasonality of transmission. We avoid these complications by investigating only non-seasonal settings, from which we selected 10 different resistant genotypes having a known selection coefficient and quantified their probability of establishment (see Materials and methods). By doing so, we evaluated the relationship between the selection coefficient and probability of establishment and assessed how this relationship varies across settings due to variation in the heterogeneity of parasite reproductive success.

As expected, the establishment of a mutation was more probable when its selection coefficient was high (*Figure 4*). For each treatment profile, the probability of establishment of mutations with similar selection coefficients was higher at low EIR than at high EIR (*Figure 4*), especially for mutations with a high selection coefficient. The lower probability of establishment in higher transmission settings suggests that higher transmission levels increase the heterogeneity in parasite reproductive success, reducing the chance to transmit advantageous mutations. Two factors increase the heterogeneity of parasite reproductive success in settings with a high EIR. First, in higher transmission settings, there is higher variability in the number of parasites with distinct genotypes carried by a host and which are competing for reproductive success. Thus, the greater variability of within-host competition between hosts leads to greater heterogeneity of parasite reproductive success. Second, settings with a high EIR have a large variation in the level of individual immunity. Host immunity influences parasite reproductive success by reducing parasite growth within the human host. Therefore, in high transmission settings, the greater variation of immunity leads to higher heterogeneity of parasite reproductive success and reduces the chance that the emerging mutation will be transmitted despite its advantage.

## Discussion

Understanding which disease, transmission, epidemiological, health system, and drug factors systemically drive the evolution of drug resistance is challenging. A full understanding requires vast observational data or clinical trials on a scale that is not possible or mathematical models that are sufficiently detailed to capture all these factors while remaining computationally feasible to simultaneously assess the impact of these factors. In response to this need, we updated a detailed individual-based model of malaria dynamics to include a full pharmacological (i.e. PK/PD) description of antimalarial treatments. We introduced a global sensitivity analysis approach based on emulators for computationally intensive models to systematically assess which factors jointly drive the evolution of drug-resistant parasites. As discussed below, our approach allowed us to understand the guiding principles of the evolution of drug resistance against ACTs and to explain the difference in trends observed in the GMS and in malaria endemic Africa. Improving our understanding of the factors that lead to drug resistance establishment and spread allows us to identify strategies to mitigate these dynamics and guides initial considerations for developing more sustainable malaria treatment.

Our results support the belief that evolution of resistance to ACTs begins with the establishment and spread of parasites resistant to the partner drug and once the protective effect of the partner drug is reduced, drug selection falls on the artemisinin component, and parasites then start to acquire resistance to artemisinin derivatives (e.g. *Watson et al., 2021*; *Hastings et al., 2016*). The fact that

resistance to the partner drug appears before resistance to artemisinin derivatives was supported by two points elucidated in our study. First, resistance to the partner drug depends on the period of low concentration of this drug during which only resistant parasites can multiply within the host (known as the selection window). As artemisinin derivatives are short-acting, they cannot prevent patients from being reinfected by parasites resistant to the partner drug during this selection window. Second, resistance to the partner drug was the most critical factor that enhanced establishment and spread of partial artemisinin resistance. Without resistance to the partner drug, parasites partially resistant to artemisinin could only spread at a low rate as the partner drug could still eliminate them, thereby removing their selective advantage. Our results are in line with recent molecular data which show that parasites resistant to partner drugs (piperaquine and mefloquine) were already present in the GMS before partial artemisinin resistance emerged and that the spread of resistance to artemisinin accelerated when it became linked to resistance to the partner drugs (*Amato et al., 2018*; *Hamilton et al., 2019*; *Wongsrichanalai and Meshnick, 2008*). Thus, the presence of partner drug resistance has probably facilitated the spread of resistance to artemisinin in the GMS. In contrast, in Africa, to date, only a low degree of resistance to the most commonly used partner drugs (lumefantrine and amodiaquine) are present (*WHO, 2020b*; *Ehrlich et al., 2021*), which has likely limited establishment of resistance to artemisinin derivatives. We additionally note that the evolution of drug resistance in the GMS may have been favoured by the low transmission intensity (annual EIR range approximate from less than 1 to 25 inoculations per person per year [*Edwards et al., 2019a*; *Chaumeau et al., 2018*; *Edwards et al., 2019b*]) compare to Africa where the transmission intensity is overall higher (annual EIR range from less than 1 to more than 500 inoculations per person per year [*Hay et al., 2000*; *Yamba et al., 2020*]). Similar to previous studies (*White, 1999*; *Bushman et al., 2018*; *Pongtavornpinyo et al., 2008*; *Lee et al., 2022*; *Hastings, 1997*), establishment of drug resistance in our model was more likely in low transmission settings due to the reduced level of within-host competition between genotypes, as well as population immunity.

Our results suggest that a key strategy to mitigate the evolution of partial artemisinin resistance is to ensure that the partner drug efficiently kills the partially resistant parasite. Therefore, to delay the establishment of artemisinin resistance in Africa and to mitigate the spread of partial artemisinin resistance in regions where it is already established, we should ensure that limited or no genotypes are resistant to the partner drug for first-line ACT. One approach to ensure this is to implement robust molecular surveillance of resistance markers and to specify more sustainable treatment policies, such as changing first-line ACTs upon detection of resistance or when the frequency of resistant parasites reach a threshold as recommended by the WHO (*WHO, 2020b*). Furthermore, consistent with our results, adherence should continue to be promoted, as lower treatment compliance can lead to treatment failure even in the absence of resistance to the partner drug (*WHO, 2020b*; *Siddiqui et al., 2015*; *Bruxvoort et al., 2014*).

Our results suggest that future antimalarial therapies should shorten the selection windows of long-acting partner drugs. We show that resistance to long-acting drugs is the first step in the evolution of resistance to ACTs, and it depends mainly on the length of the selection window. We confirm that the selection window strongly depends on the drug half-life, also consistent with previous studies (*White, 1999*; *Slater et al., 2017*; *Hastings et al., 2002*; *Watkins and Mosobo, 1993*; *Pongtavornpinyo et al., 2008*; *Kay and Hastings, 2015*). Consequently, reducing the half-life of the partner drug in an ACT regimen could reduce the spread of resistance. However, unless selection windows are substantially minimised or completely eliminated, the evolution of resistance would not totally be prevented (*Kay and Hastings, 2015*). Thus, a more sustainable option for ACTs would be to use TACTs. TACTs involve combining an artemisinin derivative with two long-acting drugs (*Krishna, 2019*).

If or when TACTs are to be widely used, our results emphasise that the two long-acting drugs should have matching half-lives to ensure that parasites are not exposed to residual drug concentrations of only one of the two partner drugs (noting that this is simple in principle, but more difficult in practice [*Hastings and Hodel, 2014*]). In addition, the parasite population should be devoid of parasites resistant to either of the two long-acting drugs. If resistance to one partner drug already exists in the population, the second partner drug would not be protected, and mutations conferring resistance to this second drug could be selected. However, additional forces will play a role in the evolution of resistance to drugs used in TACTs, such as if the drugs combined lead to opposite selection pressure. Additional analyses should assess which factors promote drug resistance under TACTs to guide

their development. Note that the development of new partner drugs for TACTs may be challenging because combining three drugs is likely to increase the risk of toxicity and the treatment price, and future antimalarial medicines must remain tolerated by patients and affordable (**Krishna, 2019**).

Another approach to delay the evolution of partial artemisinin resistance could focus on extending the period of action of artemisinin derivatives. In our monotherapy analysis on the spread of a genotype partially resistant to artemisinin, we found that the spread of partially resistant genotypes decreased when the drug was present in patients for a longer time, such as if it had a long half-life and there was proper treatment adherence. This result arises because partially resistant parasites are still affected by the drug (**Klonis et al., 2013**; **Sá, 2018**; **Witkowski et al., 2013**; **Ye et al., 2016**). Thus, increasing their exposure to the drug leads to higher killing and reduced spread. Increasing the exposure to artemisinin derivatives can be achieved by using the artemisinin derivative having the longest half-life and, as highlighted in other studies (**Kay et al., 2015**; **Dogovski et al., 2015**; **Khoury et al., 2020**), can be done by increasing the number of doses and days that patients receive treatment. However, it is worth noting that extending the dosage regimen will be efficient only with adequate adherence to treatment, which may be challenging to achieve in practice. Also, as artemisinin derivatives are co-administered with at least one long-acting drug, increasing the number of doses of this combination therapy would require reducing the concentration of the partner drug to prevent the partner drug from reaching toxic concentrations.

The evolution of drug resistance is a three-step process consisting of mutation, establishment, and spread. Mutation rates in malaria can be measured. Spread, quantified by the selection coefficient, is also easy to measure. However, the probability of establishment and its relation to the selection coefficient constituted a significant knowledge gap. Standard population genetic models assume that the number of secondary infections follows a Poisson distribution (**Hastings, 2004**; **Crow and Motoo, 2017**). Under this assumption, for selection coefficients lower than 0.2 (according to an informal literature review in **Hastings et al., 2020**, most selection coefficient estimates for malaria drug resistance mutations from the field fall between 0.02 and 0.12), the probability of establishment is approximately equal to twice the selection coefficient (**Hastings, 2004**; **Crow and Motoo, 2017**). However, the number of secondary malaria infections more likely follows a negative binomial distribution due to the high heterogeneity of transmission, which may substantially reduce the probability of establishment (Box 2 of **Hastings, 2004**). In this modelling study, we were uniquely able to quantify the link between selection coefficients and the probability of establishment of mutations. On average, we estimated that, for selection coefficients lower than 0.2, the probability of establishment was equal to 0.87 times the selection coefficient. Therefore, our findings suggest that the variation in the number of secondary infections of *P. falciparum* must be much greater than the Poisson distribution assumed by standard population genetics models, and this higher variation reduces the probability of establishment of emerging mutations. Note that higher heterogeneity in parasite reproductive success may exist in the real-world than as simulated in our model due to factors not captured by our model (such as geographical heterogeneity of exposure to mosquito bites). These factors may further decrease the probability of establishment (**Klein, 2014**).

As with all modelling studies, our approach has several limitations, primarily arising from constraints imposed by the model. First, our drug action model does not capture stage-specific killing effects, so we could not model parasites partially resistant to artemisinin being insensitive to the drug only during extended ring-stage (**Klonis et al., 2013**; **Sá, 2018**; **Witkowski et al., 2013**; **Ye et al., 2016**), although previous analyses suggested this would be captured by our variation in the maximum killing rate (**Hodel et al., 2016**). Nevertheless, if we modelled a reduction of the drug effect restricted to the ring-stage, we expect to obtain similar results. That is, a long half-life and high treatment adherence would increase the likelihood that the drug is present within patients during any stage other than the ring-stage, and thus the drug would kill more resistant parasites.

Second, our model did not capture the impact of artemisinin resistance on gametocytes. Previous studies have highlighted that artemisinin kills gametocytes, and patients infected with parasites partially resistant to artemisinin exhibit higher gametocyte densities than patients infected with sensitive parasites (**Ashley et al., 2014**; **Witmer et al., 2020**). We did not model the impact of artemisinin and resistance on gametocytes. This effect is likely to accelerate the spread of partial resistance. However, the relationship between the different factors reported in this study should be unchanged.

Third, our model, OpenMalaria, does not capture the recombination of *P. falciparum* parasites in mosquitoes (it does not track different genotypes in mosquitoes, and the genotype of a new infections is based on the genotype frequency in humans). In practice, this means we can only investigate the spread of resistance at one locus at a time (because if there is no genetic variability at other loci, then the lack of recombination has no impact). Our results, therefore, apply to the case when resistance is already fixed for one drug before resistance starts to spread to the second; we cannot model the simultaneous spread of resistance to two or more drugs. Moreover, the resistant genotype had a fixed degree of resistance across the simulation and could not acquire additional mutations that provide higher degrees of resistance. Nevertheless, by varying the degree of resistance in our analysis, we were able to assess the changing pattern of selection that occurred with increasing degrees of resistance. The impact of recombination when genetic variability does exist at more than one locus involved in resistance has been investigated previously by simpler genetic models whose main results are as follows. When multiple mutations are needed to confer drug resistance, recombination can slow the spread of drug resistance by separating these mutations (*Hastings, 1997*; *Dye and Williams, 1997*). Resistance to some partner drugs requires multiple mutations (*WHO, 2020b*). Partial artemisinin resistance is caused by a mutation in a single gene, but recombination may still impact its spread by separating this mutation from mutations that can minimise the fitness cost associated with resistance (*Stokes et al., 2021*). Recombination is more likely to impact the spread of resistant parasites in high transmission settings where recombination between different parasite genotypes is more likely. In addition, the impact of recombination depends on the frequencies of mutations involved in the resistant phenotype (*Dye and Williams, 1997*). When their frequencies are low, recombination will have a stronger effect as resistant parasites are more likely to recombine with sensitive parasites leading to the separation of these mutations. When their frequencies are high, the impact of recombination is reduced as resistant parasites are more likely to recombine with resistant parasites. A consequence of not including recombination is that in high transmission settings, we have probably overestimated the probability of establishment of resistant parasites that have multiple mutations involved in the drug-resistant phenotype. This means that the difference between the probabilities of establishment in low and high transmission settings is likely greater than reported here. In addition, we may have overestimated the spread of these resistant parasites when these mutations are present in low frequencies.

Lastly, to investigate the establishment of drug-resistant parasites, we modelled the emergence of mutations through importation. Consequently, our estimations represent the establishment of mutations imported into a population or mutations emerging in mosquitoes (assuming that the mosquito has only transmitted the mutated genotype and not the wild type genotype to the individual). A mutation emerging during the blood-stage within the human host may have a lower probability of establishment because sensitive parasites would be present in the host, leading to competition between them. It is still unclear whether mutations conferring drug resistance arise during the blood-stage (due to the high parasite numbers) or during the sexual stage in mosquitoes (because recombination generates many genetic variations). Nevertheless, the probabilities of establishment estimated in this study are consistent with the probabilities of establishment estimated by a previous study (*Hastings, 2004*).

In summary, our results confirm that mutations conferring malaria drug resistance are more likely to establish in low transmission settings. Our results demonstrate that the establishment and spread of resistance to artemisinin derivatives have likely been facilitated by pre-existing resistance to partner drugs. Thus, it is essential to prioritise monitoring and to limit the spread of resistance to partner drugs in current or future ACT regimens. If resistance to the partner drug is confirmed, response strategies should prioritise monitoring molecular markers and treatment failure and switching to an ACT with an effective partner drug should be considered. In addition, our results show that drug properties play an essential role in the evolution of drug resistance. Thus, the ongoing development of new antimalarial combinations should limit selection windows of partner drugs by matching half-lives, hopefully leading to longer lasting combination treatments against malaria. In the medium-term, for existing ACTs, it would be advantageous to increase the time of parasite exposure to the short-acting artemisinin derivate and/or to include a second long-acting partner drug with a matching half-life to the other long-acting partner drug (triple ACTs *Krishna, 2019*) and for which limited or no parasite resistance exists in the target population.

## Materials and methods

### Simulation model and the parameterisation of treatment profiles and resistant genotypes

#### Overview of our OpenMalaria model

Our individual-based model, OpenMalaria, simulates the dynamics of *P. falciparum* in humans and links it to a periodically forced deterministic model of *P. falciparum* in mosquitoes (***Chitnis et al., 2012***; ***Smith et al., 2006a***; ***Smith et al., 2008***). The model structure and fitting are described in detail elsewhere (***Smith et al., 2006a***; ***Smith et al., 2008***), including open-access code (https://github.com/SwissTPH/openmalaria) and documentation (https://github.com/SwissTPH/openmalaria/wiki), and a recently published manuscript provides a new calibration (***Reiker et al., 2021***). Here, we have summarised the main components of OpenMalaria and its latest developments in version 40.1, which enabled us to model the establishment and spread of drug-resistant parasites.

OpenMalaria includes multiple sub-models in which mosquito and infection events, parasite, and human attributes are updated every 5 days. A demography component maintains a constant human population size and age structure across the simulation. Multiple parasite genotypes and their initial frequency can be defined in more recent model versions. For each infection, a mechanistic model simulates the parasite dynamics within the host and incorporates innate, variant, and acquired immunity (***Molineaux et al., 2002***). The within-host model allows for concurrent infection of multiple parasite genotypes within the same host and captures indirect competition between genotypes based on host immunity, which regulates the overall parasite load. The user can specify a reduction of the within-host multiplication factors of each genotype to model a fitness cost associated with the mutation. The host's parasite density determines the symptoms and mortality of patients and diagnostic test results. The occurrence and severity of patient symptoms depend on their pyrogenic threshold, which increases (until saturation) with recent parasite exposure and decays over time (***Smith et al., 2006b***). Severe episodes of malaria occur due to a high parasite density or due to co-morbidities (***Ross et al., 2006b***). Malaria mortality can be a consequence of a severe episode or an uncomplicated episode with co-morbidity (***Ross et al., 2006b***; ***Ross and Smith, 2006c***). The model also takes into account neonatal deaths (***Ross et al., 2006b***; ***Ross and Smith, 2006c***). Immunity to asexual parasites prevents severe cases by decreasing the parasite multiplication rate within the host. Individual immunity depends on the cumulative parasite and infection exposure frequency, as well as maternal immunity in infants for several months (***Maire et al., 2006a***).

The case management component of OpenMalaria describes the use of treatment for uncomplicated and severe cases and depends on access to health services and whether patients have previously been treated for the same episode (***Tediosi et al., 2006***). The disease model includes explicit PK/PD models that capture the process whereby drugs reduce the parasite multiplication rate in treated hosts (***Bertrand and Mentré, 2008***; ***Winter and Hastings, 2011***). Pharmacodynamics parameters are parameterised individually for each genotype to allow different degrees of drug susceptibility to be modelled.

The entomological component of OpenMalaria simulates the mosquito vector feeding behaviours and tracks the infectious status of mosquitoes (***Chitnis et al., 2012***). The periodicity of this model allows seasonal patterns of transmission to be captured. The probability that a feeding mosquito becomes infected depends on the parasite density within bitten individuals (***Ross et al., 2006a***). No recombination is modelled between the different genotypes in the mosquitoes. The number of newly infected hosts depends on the simulated EIR of the vector model (***Chitnis et al., 2012***). The genotype of new infections is based on the genotype frequencies in humans from the previous five time steps (***Ross et al., 2006a***).

#### Parameterisation of the treatment profiles

This study investigated factors influencing the establishment and spread of parasites resistant to three different treatment profiles.

The first treatment profile modelled was a short-acting drug administered as monotherapy. The short-acting drug has a short half-life and a high killing efficacy, simulating artemisinin derivatives (***Figure 1A and B***). We modelled the pharmacokinetics of the short-acting drug using a one-compartment model, which is considered sufficient when modelling short-acting antimalarials (***Kay***

*et al., 2013*; *Winter and Hastings, 2011*). We varied key PK/PD parameters (half-life, EC50, Emax) in the global sensitivity analysis to assess their influence on the rate of spread of resistance. The EC50 ranged from 0.0016 to 0.009 mg/l to include the EC50 of artemether, artesunate, and dihydroarte-misinin (*Kay et al., 2013*; *Winter and Hastings, 2011*). The half-life parameter ranges represented the values for artemether, artesunate, and dihydroartemisinin used by *Kay et al., 2013*; *Winter and Hastings, 2011* (*Table 1*). Note that in *Kay et al., 2013*, the Emax of all short-acting drugs was equal to 27.6 per day. However, we varied the killing rate and included higher values to investigate its effects on the rate of spread (*Table 1*). To ensure that the short-acting drug killed the sensitive parasites efficiently for any combination of parameters, we extended the treatment course from a daily drug dose for 3 days to a daily drug dose for 6 days. Moreover, we parameterised the dosage and constant parameter values to that for dihydroartemisinin (*Appendix 1—table 1*), as it is the artemisinin deri-vate with the shortest elimination half-life and highest EC50 (*Kay et al., 2013*; *Winter and Hastings, 2011*). By doing so, we also ensured that the short-acting drug had the typical profile of an artemisinin derivative.

The second treatment profile modelled was a long-acting drug administered as monotherapy. The long-acting drug had a long half-life and a lower Emax than the short-acting drug (*Figure 1A and B*), typical of partner drugs used for ACTs. We modelled the PK of the long-acting drug with a two-compartment model, which is more typical of the clinical PK of partner drugs (*Bertrand and Mentré, 2008*). As for the short-acting drug, key PK/PD parameters (half-life, EC50, Emax, and dosage) were varied in the global sensitivity analysis. The EC50 ranged from 0.01 to 0.03 mg/l to include the EC50 of mefloquine, piperaquine, and lumefantrine used by *Kay et al., 2013*; *Winter and Hastings, 2011*. The half-life range corresponded to the value reported for mefloquine, piperaquine, and lumefantrine in *Charles et al., 2007*; *Staehli Hodel et al., 2013*; *Jullien et al., 2014*; *Karunajeewa et al., 2008*; *Maganda et al., 2015* (*Table 1*). We increased the Emax range from 3.45 per day (as reported in *Winter and Hastings, 2011*) to 5.00 per day to investigate the effect on the rate of spread (*Table 1*). We also assessed the impact of Cmax on the rate of spread for the long-acting drug because the Cmax varies between ACTs partner drugs and has a strong influence on the post-treatment killing effect of the long-acting drug (*Hastings and Hodel, 2014*). We varied drug dosage from 30mg/kg to 40 mg/kg to examine the influence of variation of Cmax on the spread rate for the long-acting drug. The lower limit of 30 mg/kg was fixed to ensure that the long-acting drug killed the sensitive geno-type efficiently for any parameter combination. The treatment course involved a daily drug dose for three consecutive days. To ensure that the long-acting drug had the profile of typical partner drugs, the values of the constant parameters were parameterised to the values of piperaquine reported in *Winter and Hastings, 2011*; *Staehli Hodel et al., 2013* (*Appendix 1—table 2*).

The last treatment profile was a combination of short- and long-acting drugs, simulating ACT. We tracked the concentration of each drug independently. We used the same models, parameter values and ranges for the two drugs as when both drugs were used as monotherapy. However, the treatment course involved a daily dose of both drugs for 3 days, as recommended by the WHO for most ACTs (*WHO, 2021*). In OpenMalaria, the killing effects of the two drugs were calculated independently and acted simultaneously on the parasites.

## Parameterisation of the drug-resistant genotypes

For each simulation, we tracked two genotypes, one drug-resistant and one drug-sensitive. We inves-tigated the spread of resistant parasites with different degrees of resistance (*Table 1*). We modelled the phenotype of drug resistance and the degree of resistance differently for each drug profile.

Previous studies have shown that parasites partially resistant to artemisinin exhibit an extended ring-stage during which they are not sensitive to artemisinin (even at high drug concentrations) but remain sensitive to the drug during other stages of the blood replication cycle (*Klonis et al., 2013*; *Wang et al., 2017*; *Sá, 2018*; *Witkowski et al., 2013*; *Ye et al., 2016*). OpenMalaria does not model the specific drug-killing effect for the different steps of the blood-stage. As in *Lohy Das et al., 2017*; *Lohy Das et al., 2018*, we assumed that parasites resistant to the short-acting drug had a reduced Emax compared with sensitive ones (*Figure 1B*). This assumption captured the fact that, overall, the short-acting drug killed fewer resistant parasites than sensitive ones at any drug concentration because they are not sensitive to artemisinin during the ring-stage and that this stage-specific effect is best incorporated into PK/PD modelling by variation in Emax (*Hodel et al., 2016*).

Previous studies reported that parasites resistant to long-acting drugs typically have an increased EC50 (*Chaorattanakawee et al., 2016*; *Chaorattanakawee et al., 2015*; *Tahita et al., 2015*). Thus, as in other models, we defined parasites resistant to the long-acting drug to have a higher EC50 than the sensitive ones (*Figure 1B*; *Kay et al., 2013*; *Winter and Hastings, 2011*). With an increased EC50, the resistant parasites were less susceptible to the drug at low drug concentrations. Thus, these resistant genotypes were more likely to survive drug treatment and are more likely to successfully infect new hosts with higher residual drug concentrations (*Kay and Hastings, 2015*).

Considering short- and long-acting drugs in combination, the resistant genotype was resistant to the short-acting drug. But in the global sensitivity analysis, both the sensitive and resistant genotypes could have some degree of resistance to the long-acting drug. The decreased susceptibility to the long-acting drug was the same for both sensitive and resistant genotypes, meaning that we assumed the two genotypes differed only in one mutation, which conferred resistance to the short-acting drug. This assumption allowed us to ignore the effect of recombination in the mosquitoes. In effect, this assumed that the allele defining the degree of resistance to the long-acting drug was fixed in the population.

## Approach to identify the key drivers of the spread of drug-resistant parasites

Through global sensitivity analyses, we quantified how the factors in *Table 1* influenced the spread of drug-resistant parasites for each treatment profile. First, we estimated the effect of each factor in a non-seasonal setting with a population fully adherent to treatment. Based on these results, we identified specific settings for further analysis, which used constrained sensitivity analyses to investigate the impact of varying drug properties and fitness costs in a fixed set of settings (i.e. in low and high transmission settings, with low and high treatment levels of monotherapy or combination therapy) and with a fixed degree of resistance. In these additional constrained sensitivity analyses, we also investigated the effect of drivers in seasonal transmission settings (based on the seasonality pattern of a setting in Tanzania [*Maire et al., 2006b*, *Appendix 1—figure 2*]) and where populations adhere to either 100 or 67% of treatment doses.

Due to the number of factors investigated, each global sensitivity analysis required a large number of simulations (see details below) that is computationally infeasible for detailed individual-based models. Therefore, we trained an HGP (*Binois and Gramacy, 2021*) on a limited set of OpenMalaria simulations (3500–11,500 simulations). We then used the trained emulator to predict the output of OpenMalaria for a large number of simulations and used these outputs to perform the global sensitivity analysis (*Figure 1C*), adapting a similar approach to *Golumbeanu et al., 2022* and *Reiker et al., 2021*. Our approach involved: (i) randomly sampling combinations of parameters; (ii) simulating and estimating the rate of spread of the resistant genotype for each parameter combination in OpenMalaria; (iii) training an HGP to learn the relationship between the input (for the different drivers) and output (the rate of spread) with iterative improvements to fitting through adaptive sampling; and (iv) performing a global sensitivity analysis based on the Sobol variance decomposition using the trained emulator (*Kilian et al., 2000*). Each step of the workflow is detailed below.

### Random sample combinations of parameters

We randomly sampled 250 different parameter combinations from the parameter space shown in *Table 1* using an LHS algorithm (*Gramacy, 2007*). The parameter ranges were defined as follows. We defined the ranges for the properties of the short-acting drug and the long-acting drug to include the typical parameter values of artemisinin derivatives and long-acting partner drugs, respectively (*Kay et al., 2013*; *Winter and Hastings, 2011*; *Charles et al., 2007*; *Staehli Hodel et al., 2013*; *Jullien et al., 2014*; *Karunajeewa et al., 2008*; *Maganda et al., 2015*). The range of the degree of resistance captured the spread of drug-resistant parasites, which vary from fully sensitive to having almost no drug sensitivity. The fitness costs were extracted from studies investigating the decline of chloroquine-resistant parasites after the drug pressure was removed (*Kublin et al., 2003*; *Mita et al., 2003*). The variation in annual EIR captured settings with low transmission to those with high transmission. The range of access to treatment captured settings with low to high level of access to treatment. The variation in the diagnostic detection limit captured the range of sensitivity of typical diagnostics used for malaria (such as rapid diagnostic test, microscopy, and PCR) (*Kilian et al., 2000*; *Murray et al., 2008*).

## Simulate and estimate the rate of spread of the drug-resistant genotype

We quantified the rate of spread through the selection coefficient, a measure widely used in population genetics to assess the strength of selection on a genotype (*Hastings et al., 2020*). The selection coefficient is the rate at which the logit of the resistant genotype frequency increases each parasite generation and should be linear throughout the spread (*Hastings et al., 2020*). Population genetics theory often assumes an infinite population size to remove stochastic fluctuation of the allele frequency also called genetic drift (*Hastings et al., 2020*). However, in our model the parasite population size is finite, so stochastic fluctuations are present. Thus, we should avoid estimating the selection coefficient when there is a low frequency of the resistant genotype (from a small human population size, a low EIR, and a small initial frequency of the resistant genotype) because the resistant genotype may become extinct due to the stochastic fluctuation. In addition, the effects of genetic drift that occurs when a genotype is present at a low frequency may cause non-linearity during resistance spread which may obscure the estimation of the selection coefficient (*Hastings et al., 2020*).

Following the approach described in *Hastings et al., 2020*, we assumed an initial percentage of infected humans carrying the resistant genotype of 50%. A high initial percentage minimises the impact of random fluctuation on our estimation, and the subsequent risk of extinction, without affecting our estimate because the selection coefficient was not frequency-dependent (*Appendix 1—figure 3*). We simulated the spread of resistant parasites in a human population of 100,000 individuals with an age structure typical of some countries in Africa (17.7% of people under 5 years of age) (*Ekström et al., 2016*). We ran each parameter combination on five stochastic realisations. The simulation started with a burn-in period of 100 years to reach the expected level of immunity in the population and an additional 30 years to reach EIR equilibrium (*Appendix 1—figure 4*). Both genotypes were sensitive to the drug during this period, so the percentage of infected humans carrying the resistant genotype remained stable. After the burn-in period, we introduced the fitness cost and the drug for which the resistant genotype had reduced sensitivity. We then estimated the selection coefficient, *s*, as,

$$s = \frac{1}{t}\left(ln\left(\frac{p(t+1)}{1-p(t+1)}\right) - ln\left(\frac{p(1)}{1-p(1)}\right)\right) = \frac{1}{12}\left(ln\left(\frac{p(13)}{1-p(13)}\right) - ln\left(\frac{p(1)}{1-p(1)}\right)\right),$$

where $p(t)$ is the frequency of the resistant genotype in inoculations (the number of inoculations carrying the resistant genotype divided by the total number of inoculations resistant and sensitive genotypes), $t$ is the number of parasite generations after introducing the new drug at $t=0$. We assumed that a parasite generation is 2 months (60 days) as in *Hastings et al., 2020*. We started the regression at one parasite generation after introducing the new drug (at 60 days). We stopped the regression 12 generations later, at 720 days, because, as shown in *Hastings et al., 2020*, it was computationally convenient and returned stable selection coefficient estimates. The regression was stopped sooner if the frequency of inoculations carrying the resistant genotype was higher than 90% or lower than 30% to prevent tracking a small number of a single genotype for which genetic drift is strong. In seasonal settings, the rate of spread of the resistant genotype varied throughout the year. Consequently, we estimated the selection coefficient using a moving average of the frequency of the resistant genotype in inoculations (*Appendix 1—figure 5*). This method prevented biasing the selection coefficient according to the period included in the regression.

Once the selection coefficient was estimated, it could be converted to the number of parasite generations needed for the frequency of the resistant genotype in inoculations to increase from *p(1)* to *p(t)*,

$$t = \frac{1}{s}\left(ln\left(\frac{p(t+1)}{1-p(t+1)}\right) - ln\left(\frac{p(1)}{1-p(1)}\right)\right).$$

We could then convert the number of parasite generations to time in years, a more relevant public health measure than the selection coefficient itself.

## Train the emulator and improve its accuracy

We randomly split our data into a training dataset containing 80% of simulations and a test dataset containing 20% of simulations. We trained the HGP on the training dataset using the function mleHetGPfrom the R package 'hetGP' (*Binois and Gramacy, 2021*). We chose to use HGP as it was successfully used in two previous studies that performed global sensitivity analyses of OpenMalaria

(*Reiker et al., 2021*; *Golumbeanu et al., 2022*). In addition, *Reiker et al., 2021* tested different emulators and found that HGP provided the best fit with a limited number of simulations (analysis not shown in the published study). To assess the accuracy of the emulator, for the test dataset we assessed the correlation coefficient and root mean squared error between selection coefficients estimated with the emulator and selection coefficients estimated using OpenMalaria. We iteratively improved the accuracy of our emulator through adaptive sampling. Adaptive sampling involved resampling 100 parameter combinations in the parameter space where we were less confident (higher variation) in the HGP prediction and repeating the entire process until the emulator had a satisfactory level of accuracy. The satisfactory level of accuracy was defined based on the correlation coefficient and the root means squared error between the estimated selection coefficient and expected selection coefficient for the test dataset (*Appendix 1—figures 6–12*).

### Global sensitivity analysis

Using the emulator, we undertook global sensitivity analyses using Sobol's method (*Sobol, 2001*). This method attributed fractions of the selection coefficient variance to each input (*Sobol, 2001*). To do this, we first generated two random datasets with a sample size of 100,000 using an LHS algorithm (*Gramacy, 2007*) that sampled within the parameter ranges of *Table 1*. When then estimated selection coefficients for these datasets with the trained emulators. Note that without emulators, we would have to run these simulations in OpenMalaria, which would not have been feasible due to computational requirements. We then used the function soboljansen from the R package 'sensitivity' to perform the global sensitivity analysis with 150,000 bootstrap replicates and the two datasets (*Cheng et al., 2021*). With this function, we estimated first-order and total Sobol' indices simultaneously. The first-order indices represent contributions of each parameter's main effect to the model output variance. The total effect represents the contribution of each parameter to the model output variance considering their interactions with other factors. We report only the first-order indices in the Results section because we did not observe many interactions between these factors. Some parameters supported the spread of resistance (increased the selection coefficient), whilst others hindered the spread (decreased the selection coefficient). To visualise the direction of the effect of each parameter, we calculated the 25th, 50th, and 75th quantiles of the estimated selection coefficient of the two random datasets over the corresponding parameter ranges.

## Establishment of drug resistance

As explained in the Introduction, the establishment of resistant mutations is a stochastic process that depends on the selection coefficient of the mutation and the heterogeneity of parasites reproductive success in the setting, which in turn depends on the transmission level and the health system strength (*zur Wiesch et al., 2011*; *Hastings, 2004*; *Hastings et al., 2020*; *Hastings and Mackinnon, 1998*; *Klein, 2014*). Estimating the probability of establishment requires running many stochastic realisations due to the stochasticity of this step. To be more computationally efficient, we assessed the probability of establishment of a subset of 10 resistant genotypes with a known selection coefficient per setting and treatment profile. Based on the observed relationships between the selection coefficient and the probability of establishment for each treatment profile and setting, we could then extrapolate the probability of establishment of any mutations having a known selection coefficient.

   To estimate the probability of establishment, we modelled the emergence of resistant genotypes in a fully susceptible parasite population. We used the approach described in *Hastings et al., 2020*, in which resistant infections were imported into the population at a low rate. In OpenMalaria, imported infections have the same frequencies of genotypes as in initialisation, thus we cannot import only resistant infections. Therefore, to import resistant infections in a population infected only by sensitive parasites, we followed the step described below (*Appendix 1—figure 13*). We first defined a 50% frequency of resistant parasites in infected humans. The simulation started with a burn-in phase of 100 years, during which both genotypes were sensitive to treatment. This meant that the frequency of the resistant parasites was stable (at 50%). In the second phase, we introduced a drug to which resistant parasites were hypersensitive (the drug EC50 was 100 times lower in the resistant genotype than the sensitive one). The second phase ran for 100 years, and once complete, the parasite population was fully susceptible. In the third phase, we imported new infections at a rate low enough to ensure that the previously imported resistant genotype either established or went extinct before a

new resistant infection was imported (Appendix 1: section 5.1). The third phase ran until the resistant genotype established (frequency of the resistant genotype in infected humans is equal to 50%).

The probability of establishment, $P_e$, can be estimated based on the average number of resistant infections that are imported until the resistant genotype establishes, $N_e$, as follows (the probability of a successful event can be estimated as one divided by the mean number of independent trials required to achieve the first success [*Dekking et al., 2005*]),

$$P_e = \frac{1}{N_e}.$$

We simulated 300 stochastic realisations, $R$, and estimated $P_e$, as,

$$P_e = \frac{1}{N_e} = \frac{1}{\left(\sum_{j=1}^{R} N_{m,j}\right)/R} = \frac{R}{\sum_{j=1}^{R} N_{m,j}},$$

where $N_{m,j}$ is the number of imported resistant infections until the resistant genotype established in run $j$. Re-arranging the formula shows that $P_e$ is equal to the number of resistant genotypes established in all stochastic realisations (this number is equal to $R$ as only one resistant genotype established per stochastic realisation) divided by the total number of resistant infections imported into all stochastic realisations (includes resistant genotypes that became extinct and established). We estimated the 95% confined intervals of $P_e$ (Wilson methods [*Dekking et al., 2005*]), as,

$$\left( P_e - 1.96 \sqrt{\frac{P_e(1-P_e)}{\sum_{j=1}^{R} N_{m,j}}} , P_e + 1.96 \sqrt{\frac{P_e(1-P_e)}{\sum_{j=1}^{R} N_{m,j}}} \right).$$

Note that in each stochastic realisation, we estimated $N_m$, as,

$$N_m = t_e N_i,$$

where $t_e$ is defined as the last time that the number of infections with a resistant genotype was equal to zero, that is the time (in years) until the arrival of the imported resistant infection that led to the successful establishment of the resistant genotype. $N_i$ is the number of imported resistant infections per year. Note that OpenMalaria specifies the number of imported infections, $V$, in numbers of imported infections per 1000 people per year, and half of the imported infections were sensitive. Thus, the number of imported resistant infections that occurred until one established can be estimated as,

$$N_m = t_e \left( \frac{NV}{2(1000)} \right) = 5t_e V,$$

where $N$ is the human population size. We simulated a population size of 10,000 individuals to increase computational feasibility for the large number of simulations required for our extensive global sensitivity analyses. A larger population was unnecessary, as the population size does not influence the probability of establishment unless it is extremely small (*Waxman, 2011*). This was not the case in our simulation, which had a minimum of 3018 infections in the low transmission setting.

## Data and software availability

We did not use individual participant-level data. Parameters values used in the model were informed from the literature as referred to in the main text or the Appendix. The source code for OpenMalaria was developed using the C++language and is available at https://github.com/SwissTPH/openmalaria (*Thüring et al., 2022*)and a documentation is available at https://github.com/SwissTPH/openmalaria/wiki. The analysis script was developed using the R software and is available at https://zenodo.org/badge/latestdoi/458217287. All data and codes used to produce the figures are available at https://zenodo.org/badge/latestdoi/458226427. In addition, the data used to produce the figure are included in the manuscript.

## Acknowledgements

We sincerely acknowledge the reviewers of this manuscript who strengthened the work and interpretation. We also thank the members of the Disease Modelling Unit at the Swiss Tropical and Public Health Institute and Dr Raman Sharma from Liverpool School of Tropical Medicine for their inputs and discussions. Simulations were performed on the scientific computing core facility, sciCORE, at the University of Basel (http://scicore.unibas.ch/). This research was funded under the Swiss National Science Foundation Professorship of Melissa Penny (PP00P3_170702). Tamsin Lee also received funding from a Marie Curie Individual Fellowship (839121, Horizon 2020). The funding sources were not involved in the study design, data analysis, results interpretation, and the decision to submit the work for publication.

## Additional information

### Funding

| Funder | Grant reference number | Author |
|---|---|---|
| Schweizerischer Nationalfonds zur Förderung der Wissenschaftlichen Forschung | Professorship PP00P3_170702 | Melissa A Penny |
| Horizon 2020 Framework Programme | Individual Fellowship 839121 | Tamsin Lee |

The funders had no role in study design, data collection and interpretation, or the decision to submit the work for publication.

### Author contributions

Thiery Masserey, Data curation, Formal analysis, Investigation, Methodology, Software, Validation, Visualization, Writing - original draft, Writing – review and editing; Tamsin Lee, Investigation, Writing - original draft, Writing – review and editing; Monica Golumbeanu, Andrew J Shattock, Methodology, Software, Writing – review and editing; Sherrie L Kelly, Writing – review and editing; Ian M Hastings, Investigation, Methodology, Writing – review and editing; Melissa A Penny, Conceptualization, Funding acquisition, Investigation, Methodology, Project administration, Supervision, Validation, Writing - original draft, Writing – review and editing

### Author ORCIDs

Thiery Masserey http://orcid.org/0000-0003-2482-463X
Tamsin Lee http://orcid.org/0000-0001-6684-3615
Ian M Hastings http://orcid.org/0000-0002-1332-742X
Melissa A Penny http://orcid.org/0000-0002-4972-593X

### Decision letter and Author response

Decision letter https://doi.org/10.7554/eLife.77634.sa1
Author response https://doi.org/10.7554/eLife.77634.sa2

## Additional files

### Supplementary files

- Appendix 1—figure 1—source data 1. Related to *Appendix 1—figure 1*.
- Appendix 1—figure 6—source data 1. Related to *Appendix 1—figure 6*.
- Appendix 1—figure 7—source data 1. Related to *Appendix 1—figure 7*.
- Appendix 1—figure 8—source data 1. Related to *Appendix 1—figure 8*.
- Appendix 1—figure 9—source data 1. Related to *Appendix 1—figure 9*.
- Appendix 1—figure 10—source data 1. Related to *Appendix 1—figure 10*.
- Appendix 1—figure 11—source data 1. Related to *Appendix 1—figure 11*.

- Appendix 1—figure 12—source data 1. Related to *Appendix 1—figure 12*.
- Transparent reporting form

### Data availability

All data and code used to produce the figures are available at https://zenodo.org/badge/latestdoi/458226427. In addition, the data used to produce the figure are included in the manuscript as source data files. In addition, the code used to run the simulations and perform the analyses can be found at https://zenodo.org/badge/latestdoi/458217287. The individual-based model of malaria transmission and epidemiology used in the study has an open-access code (https://github.com/SwissTPH/openmalaria) and documentation (https://github.com/SwissTPH/openmalaria/wiki).

The following datasets were generated:

| Author(s) | Year | Dataset title | Dataset URL | Database and Identifier |
|---|---|---|---|---|
| Masserey T, Lee T, Golumbeanu M, Shattock AJ, Kelly SL, Hastings IM, Penny MA | 2022 | ThieryM95/Drug_resistance_data_and_visualisation: V1.0.3 | https://doi.org/10.5281/zenodo.6622890 | Zenodo, 10.5281/zenodo.6622890 |
| Masserey T, Lee T, Golumbeanu M, Shattock AJ, Kelly SL, Hastings IM, Penny MA | 2022 | ThieryM95/Drug_resistance_workflow | https://doi.org/10.5281/zenodo.6046967 | Zenodo, 10.5281/zenodo.6046967 |

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

## Appendix 1

### 1. Supplementary results

#### 1.1 The benefit of combination therapy

We illustrated the benefit of combination therapy by assessing how the degree of resistance to the long-acting drug influenced (i) the time taken for mutations conferring different degrees of resistance to the short-acting drug to spread from 1 to 25% of inoculations carrying the resistant genotype, *T25* (*Appendix 1—figure 1*, first y-axis) and (ii) their probability of establishment (*Appendix 1—figure 1*, second y-axis). Both the *T25* and the probabilities of establishment were estimated based on selection coefficients estimated using the fitted emulators. To illustrate the impact of the transmission intensity on the two measurements, we estimated their values in low and high transmission levels. Note that, as discussed in the Results section, the relation between the selection coefficient and the probability of establishment changes slightly with the transmission level (*Figure 4* of main text). In our example, the short-acting drug had the drug profile of dihydroartemisinin and the long-acting drug of piperaquine. We set the level of access to treatment to 100%, assumed no fitness cost, the transmission was perennial, and the population adhered to treatment fully.

In a low transmission setting, in a parasite population fully susceptible to the long-acting drug, parasites resistant to the short-acting drug had a low probability of establishment and required many years to spread from 1% to 25% of inoculations carrying the resistant genotype. For example, a mutation with a low (3.5-fold decrease in Emax) or high (13.5-fold decrease in Emax) degree of resistance to the short-acting drug had a probability of 0.016 or 0.034, respectively, to establish in the population and required 40.3 years or 17.7 years, respectively, to spread from 1 to 25% of inoculations carrying the resistant genotype (*Appendix 1—figure 1*). The probability of establishment and *T25* decreased tremendously with increased degrees of resistance of both genotypes to the long-acting drug (*Appendix 1—figure 1*). When the parasite population had a high degree of resistance to the long-acting drug (degree of resistance of 13.5), the probability of establishment increased to more than 1/10 and the *T25* was reduced to approximately 3 years, independent of the degree of resistance to the short-acting drug (*Appendix 1—figure 1*). These results confirm that resistance to partner drugs facilitates the establishment and spread of partial artemisinin resistance.

In high transmission settings, higher degrees of resistance to the long-acting drug also accelerated the establishment and spread of parasites resistant to the short-acting drug (*Appendix 1—figure 1*). However, the probability of establishment and the rate of spread were consistently lower in high transmission settings compared with low transmission settings (*Appendix 1—figure 1*). These results agree with our observations that higher levels of within-host competition and immunity minimise the establishment and spread of resistance to artemisinin in high transmission settings.

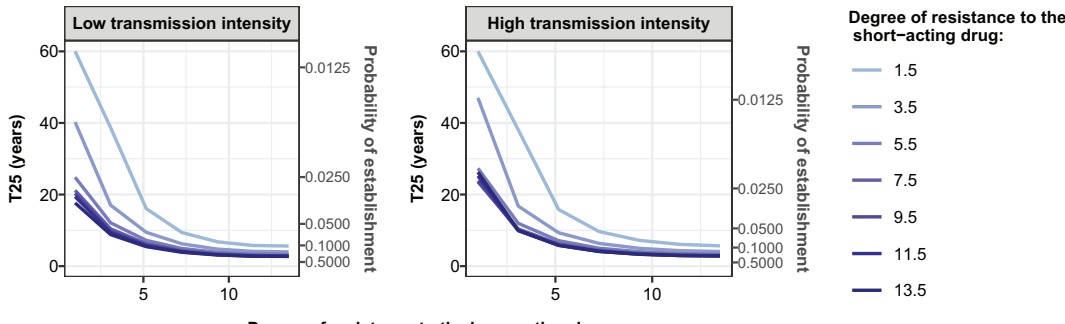

**Appendix 1—figure 1.** Illustration of the benefit of combination therapy on the evolution of drug resistance as time to 25% frequency of resistant genotypes. We estimated the probability of establishment and the time needed for parasites resistant to the short-acting drug to spread from 1 to 25% of inoculations carrying the resistant genotype, *T25,* for multiple degrees of resistance of the resistant genotype to the short-acting drug (Emax shift) and multiple degrees of resistance of both genotypes to the long-acting drug (EC50 shift). *We assumed* the short-acting drug has a similar drug profile of dihydroartemisinin and the long-acting drug of piperaquine. We assumed a level of access to treatment of 100%. The population fully adhered to treatment. The resistant parasites had no fitness cost. The transmission intensity was equal to 5 (low transmission intensity) or 500 (high transmission intensity) inoculations per person per year (reflected low to very high transmission). The transmission was perennial.

The online version of this article includes the following source data for appendix 1—figure 1:

- **Appendix 1—figure 1—source data 1.** Related to *Appendix 1—figure 1*.

## 2. Details on the parameterisation of OpenMalaria

**Appendix 1—table 1.** Pharmacokinetics (PK) and pharmacodynamics (PD) parameter values for the short-acting drug that were kept constant throughout the sensitivity analyses.

| Component | Parameter | Value | Reference |
|---|---|---|---|
| | Volume distribution (l/kg) | 1.49 | *Kay et al., 2013* |
| PK | Treatment dosage (mg/kg) | 4.00 | *Kay et al., 2013*; *Winter and Hastings, 2011* |
| PD | Slope of the effect curve | 4.00 | *Kay et al., 2013*; *Winter and Hastings, 2011* |

**Appendix 1—table 2.** Pharmacokinetics (PK) and pharmacodynamics (PD) parameter values for the long-acting drug that were kept constant throughout the sensitivity analyses.

| Component | Parameter | Value | Reference |
|---|---|---|---|
| | Absorption rate (per day) | 11.16 | *Staehli Hodel et al., 2013* |
| | Rate at which the drug moves from the central compartment to the peripheral compartment (per day) | 8.46 | *Staehli Hodel et al., 2013* |
| | Rate at which the drug moves from the peripheral compartment to the central compartment (per day) | 3.30 | *Staehli Hodel et al., 2013* |
| PK | Volume distribution (l/kg) | 173.00 | *Staehli Hodel et al., 2013* |
| PD | Slope of the effect curve | 6.00 | *Winter and Hastings, 2011*; *Staehli Hodel et al., 2013* |

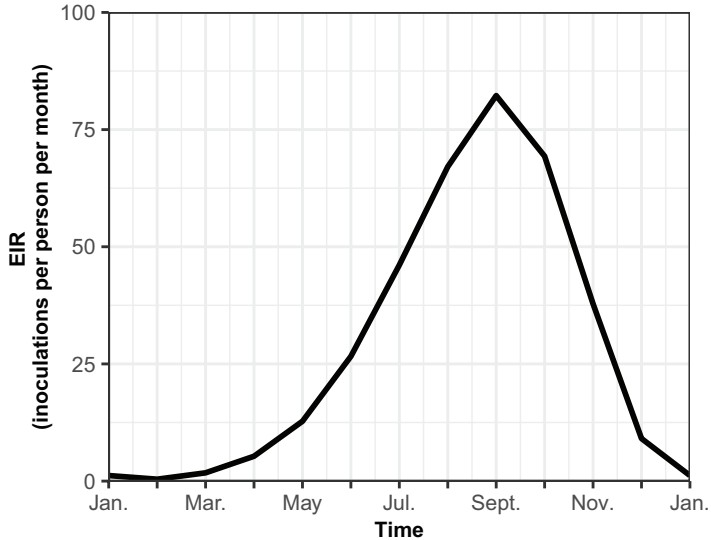

**Appendix 1—figure 2.** The seasonal transmission of malaria. Entomological inoculation rate (EIR) (inoculations per person per month) plotted for a year in our assumed seasonal setting of malaria transmission, based on field studies conducted in Tanzania (*Maire et al., 2006b*). Here the total EIR is 360 inoculations per person per year.

## 3. Estimation of the selection coefficient

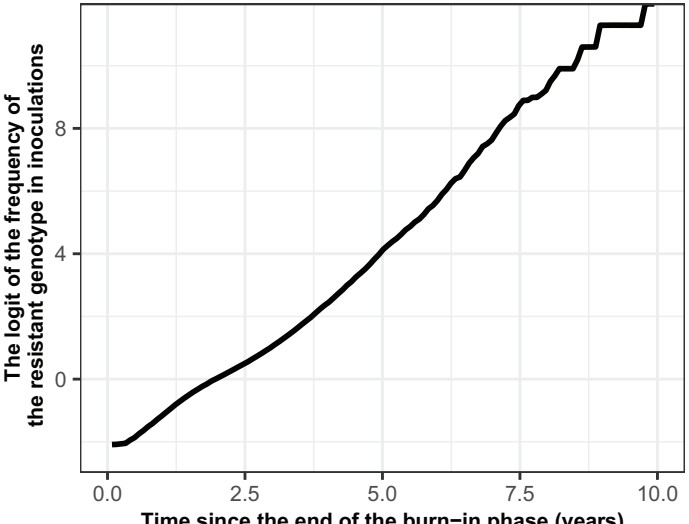

**Appendix 1—figure 3.** Proof that the selection coefficient is not frequency-dependent in OpenMalaria. The figure illustrates the logit of the frequency of the resistant genotype over time when the initial frequency of infected humans carrying the resistant genotype was 5%. The selection coefficient (slope of the logistic regression) was less stable after 6 years because the percentage of inoculations carrying the sensitive genotype was lower than 0.5%. Thus, the influence of stochastic processes was strong.

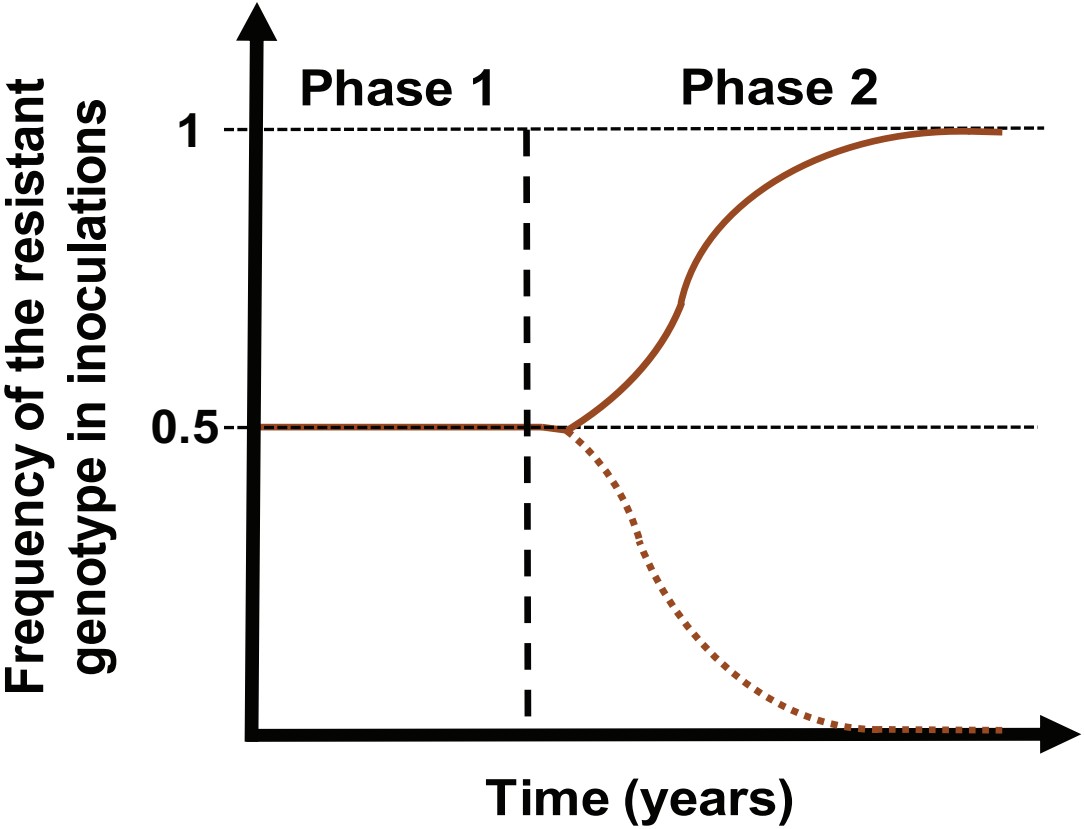

**Appendix 1—figure 4.** Schematic illustration of typical simulations run in OpenMalaria to estimate the rate of spread of a drug-resistant genotype. The brown line represents the frequency of the resistant genotypes inoculations. The solid line illustrates a simulation in which the resistant genotype spreads in the population (selection coefficient above 0). The dotted line illustrates a simulation in which the resistant genotype did not spread in the population (selection coefficient below 0). Illustrative phase 1 represents the burn-in phase. The vertical dashed black line highlights when we introduced the fitness cost and the drug for which the resistant genotype had reduced sensitivity. Phase 2 is the phase during which the rate of spread of the resistant genotype was assessed.

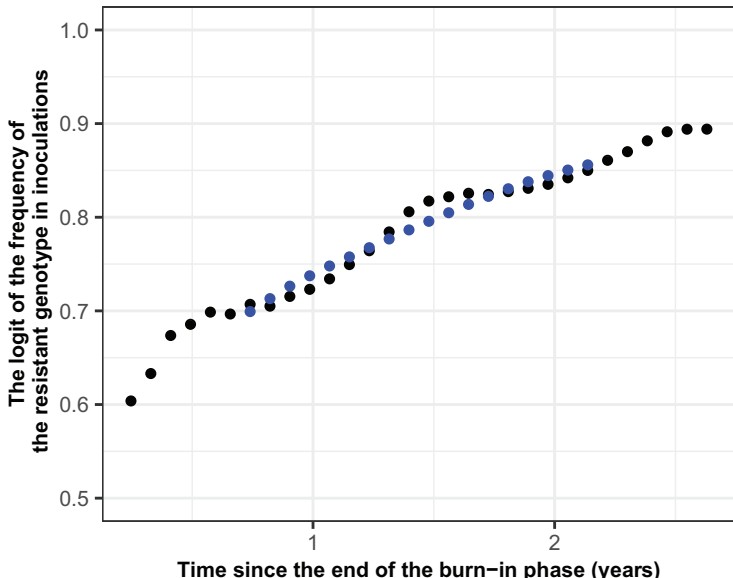

**Appendix 1—figure 5.** Illustration of the estimation of the selection coefficient in seasonal settings. The black dots represent the logit of the frequency of the resistant genotype. The blue dots represent the logit of the moving average of frequency of the resistant genotype. The moving average of a measurement at a time t included all the measurements from 6 months before time *t* and 6 months after the time *t*. Using this method, the selection coefficient (slope of the logistic regression) was constant over time.

## 4. Fit of the emulators

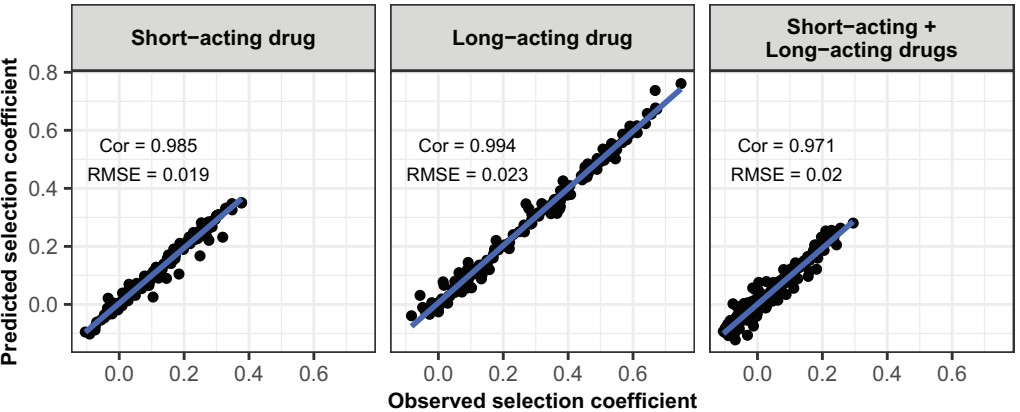

**Appendix 1—figure 6.** Accuracy of the emulators used for the global sensitivity analyses of each treatment profile. For each treatment profile, the comparison between the selection coefficients of the test dataset estimated using OpenMalaria (i.e. the observed 'true' selection coefficient) and the corresponding prediction from the emulator during the final round of adaptive sampling. 'Cor' is the Spearman correlation coefficient, 'RMSE' is the root means squared error, and the blue lines are the linear regression fits.

The online version of this article includes the following source data for appendix 1—figure 6:

• **Appendix 1—figure 6—source data 1.** Related to *Appendix 1—figure 6*.

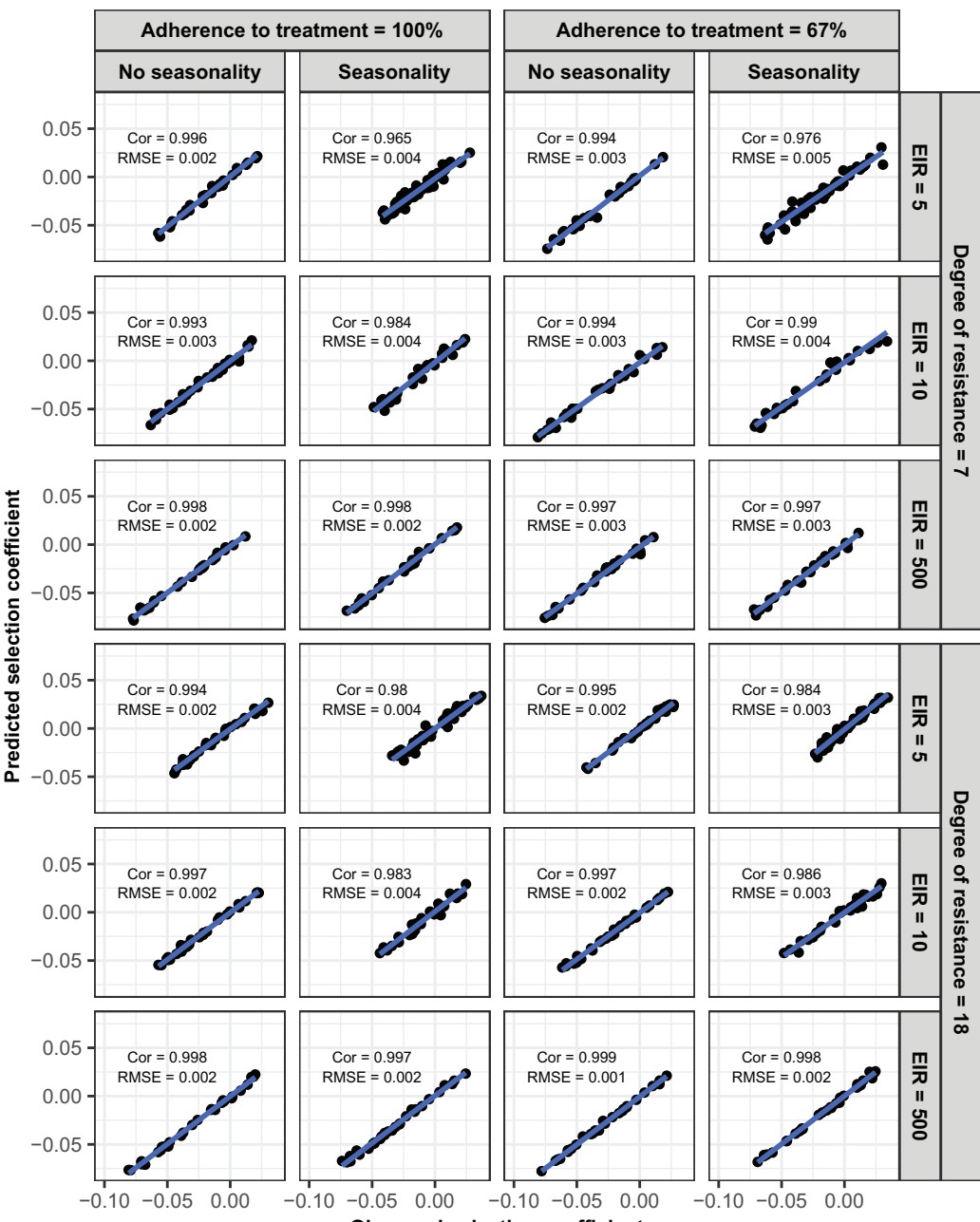

**Appendix 1—figure 7.** Accuracy of the emulators used for each constrained sensitivity analysis of the spread of a genotype resistant to the short-acting drug used in monotherapy in each setting with low access to treatment (10%). The comparison between the selection coefficients for the test dataset between the observed 'truth' from OpenMalaria, and the prediction from the emulators during the final round of adaptive sampling. The EIR is in inoculations per person per year (5, 10, and 500). The degree of resistance is the relative decrease in the Emax of the resistant genotype compared with the sensitive one. 'Cor' is the Spearman correlation coefficient, 'RMSE' is the root means squared error, and the blue lines are the linear regression fits.

The online version of this article includes the following source data for appendix 1—figure 7:

• **Appendix 1—figure 7—source data 1.** Related to *Appendix 1—figure 7*.

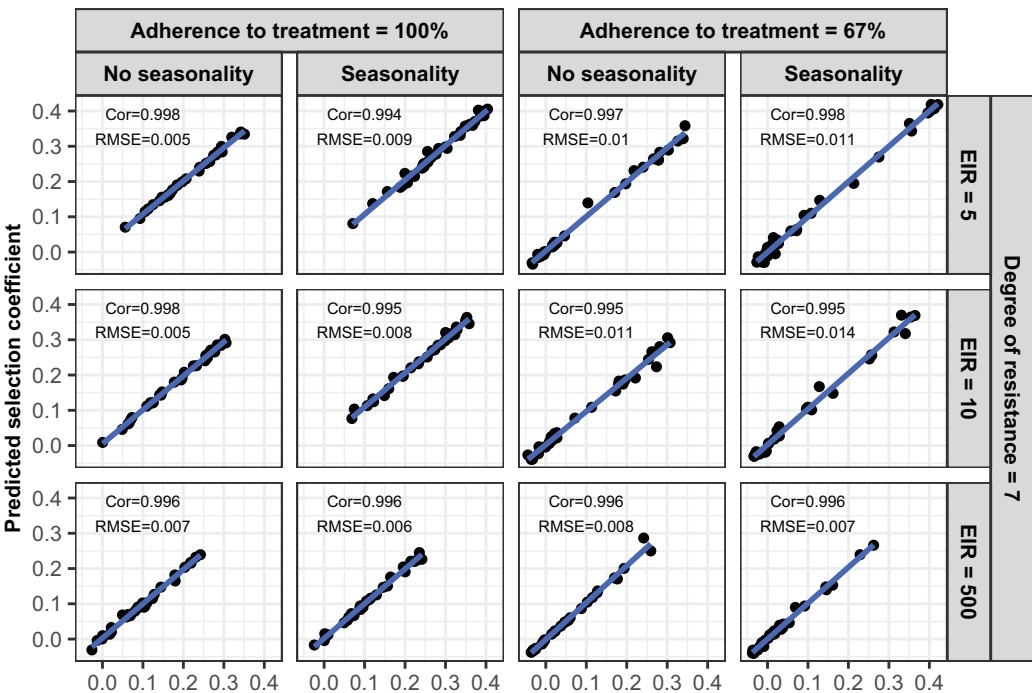

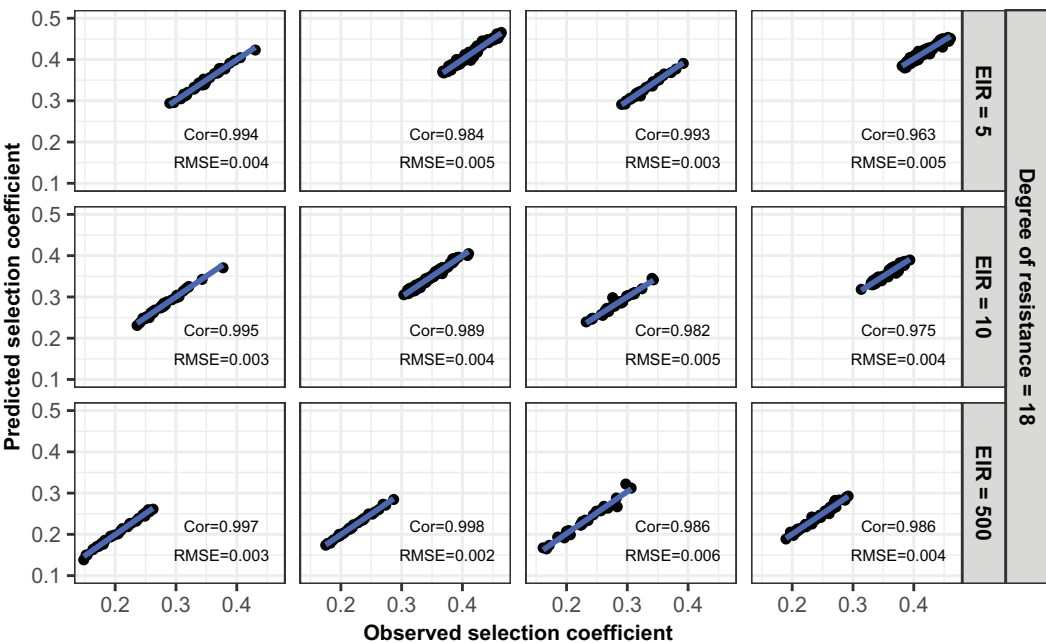

**Appendix 1—figure 8.** Accuracy of the emulators used for each constrained sensitivity analysis of the spread of a genotype resistant to the short-acting drug used in monotherapy in each setting with high access to treatment (80%). The comparison between the selection coefficients for the test dataset between the observed 'truth' from OpenMalaria, and the prediction from the emulators during the final round of adaptive sampling. The EIR is in inoculations per person per year (5, 10, or 500). The degree of resistance is the relative decrease in the Emax of the resistant genotype compared with the sensitive one. 'Cor' is the Spearman correlation coefficient, 'RMSE' is the root means squared error, and the blue lines are the linear regression fits.

The online version of this article includes the following source data for appendix 1—figure 8:

• **Appendix 1—figure 8—source data 1.** Related to *Appendix 1—figure 8*.

**Appendix 1—figure 9.** Accuracy of the emulators used for each constrained sensitivity analysis of the spread of a genotype resistant to the long-acting drug used in monotherapy in each setting with low access to treatment (10%). The comparison between the selection coefficients for the test dataset between the observed 'truth' from OpenMalaria, and the prediction from the emulators during the final round of adaptive sampling. The EIR is in inoculations per person per year (5, 10, or 500). The degree of resistance is the relative increase in the EC50 of the resistant genotype compared with the sensitive one. 'Cor' is the Spearman correlation coefficient, 'RMSE' is the root means squared error, and the blue lines are the linear regression fits.

The online version of this article includes the following source data for appendix 1—figure 9:

• **Appendix 1—figure 9—source data 1.** Related to *Appendix 1—figure 9*.

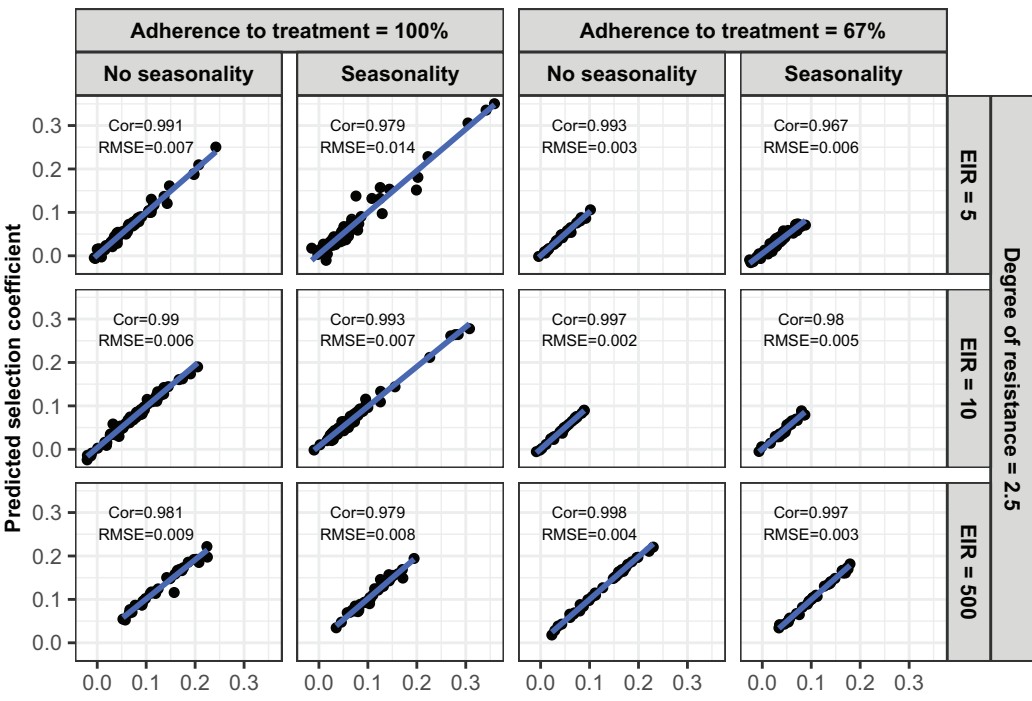

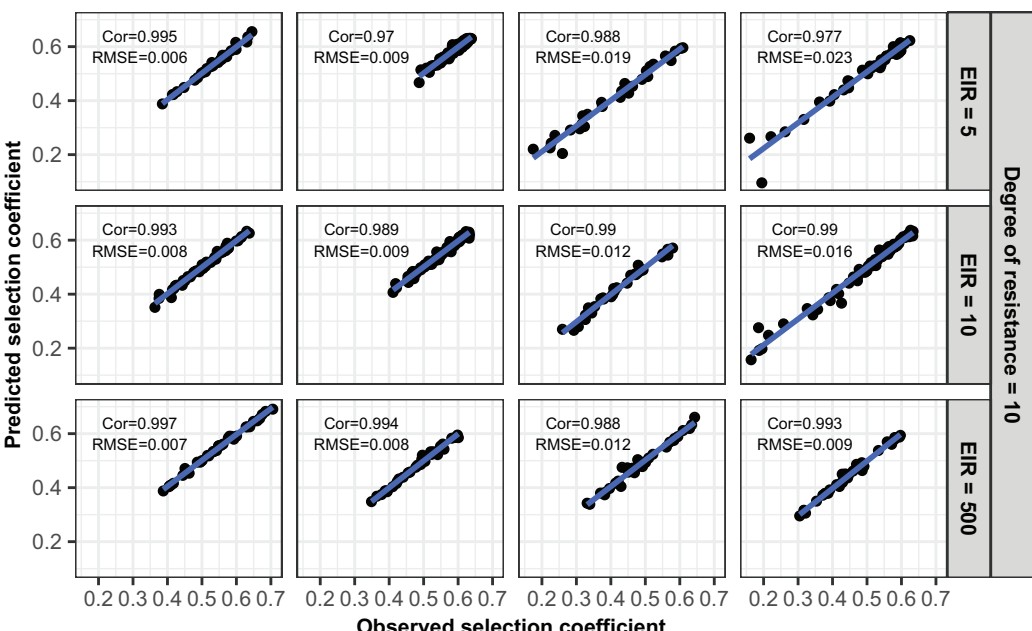

**Appendix 1—figure 10.** Accuracy of the emulators used for each constrained sensitivity analysis of the spread of a genotype resistant to the long-acting drug used in monotherapy in each setting with high access to treatment (80%). The comparison between the selection coefficients for the test dataset between the observed 'truth' from OpenMalaria, and the prediction from the emulators during the final round of adaptive sampling. The EIR is in inoculations per person per year (5, 10, or 500). The degree of resistance is the relative increase in the EC50 of the resistant genotype compared with the sensitive one. 'Cor' is the Spearman correlation coefficient, 'RMSE' is the root means squared error, and the blue lines are the linear regression fits.

The online version of this article includes the following source data for appendix 1—figure 10:

- **Appendix 1—figure 10—source data 1.** Related to *Appendix 1—figure 10*.

**Appendix 1—figure 11.** Accuracy of the emulators used for each constrained sensitivity analysis of the spread of a genotype resistant to the short-acting drug, when used in combination with the long-acting drug, in each setting with low access to treatment (10%). The comparison between the selection coefficients for the test dataset between the observed 'truth' from OpenMalaria, and the prediction from the emulators during the final round of adaptive sampling. The EIR is in inoculations per person per year (5, 10, or 500). The degree of resistance to the short-acting drug is the relative decrease in the Emax of the resistant genotype compared with the sensitive one. 'Cor' is the Spearman correlation coefficient, 'RMSE' is the root means squared error, and the blue lines are the linear regression fits.

The online version of this article includes the following source data for appendix 1—figure 11:

• **Appendix 1—figure 11—source data 1.** Related to *Appendix 1—figure 11*.

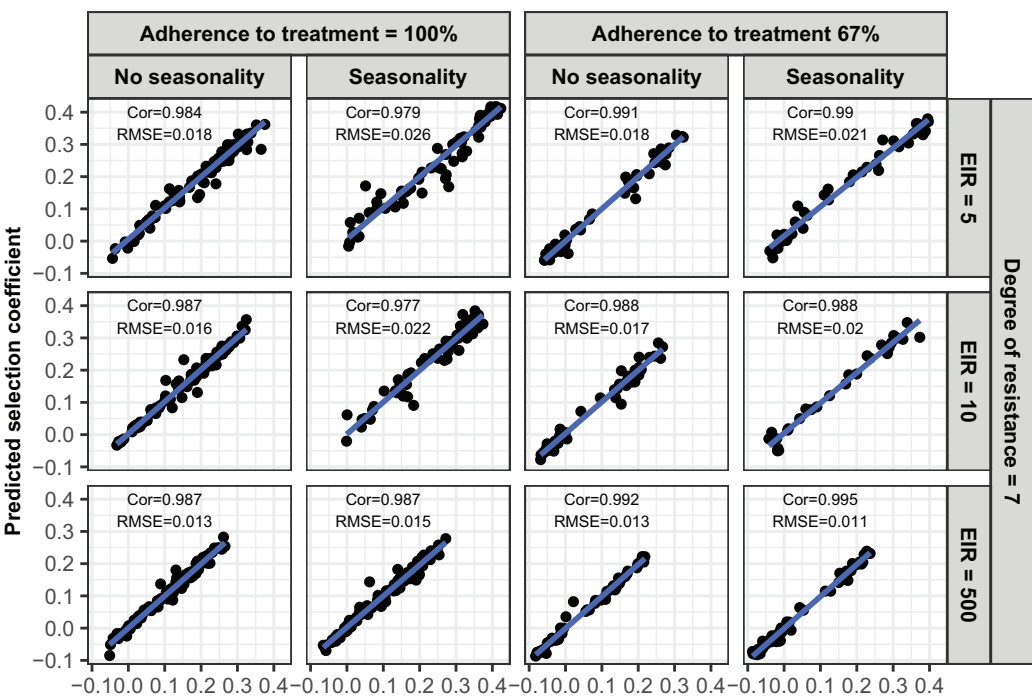

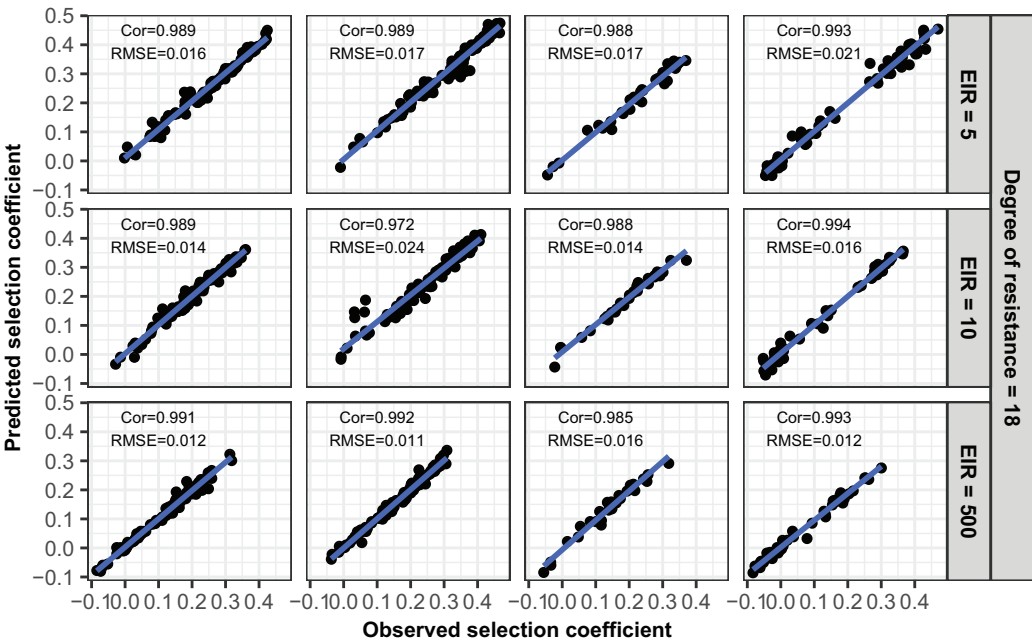

**Appendix 1—figure 12.** Accuracy of the emulators used for each constrained sensitivity analysis of the spread of a genotype resistant to the short-acting drug, when used in combination with the long-acting drug, in each setting with high access to treatment (80%). The comparison between the selection coefficients for the test dataset between the observed 'truth' from OpenMalaria, and the prediction from the emulators during the final round of adaptive sampling. The EIR is in inoculations per person per year (5, 10, or 500). The degree of resistance to the
*Appendix 1—figure 12 continued on next page*

short-acting drug is the relative decrease in the Emax of the resistant genotype compared with the sensitive one. 'Cor' is the Spearman correlation coefficient, 'RMSE' is the root means squared error, and the blue lines are the linear regression fits.

The online version of this article includes the following source data for appendix 1—figure 12:

- **Appendix 1—figure 12—source data 1.** Related to *Appendix 1—figure 12*.

## 5. Probability of establishment

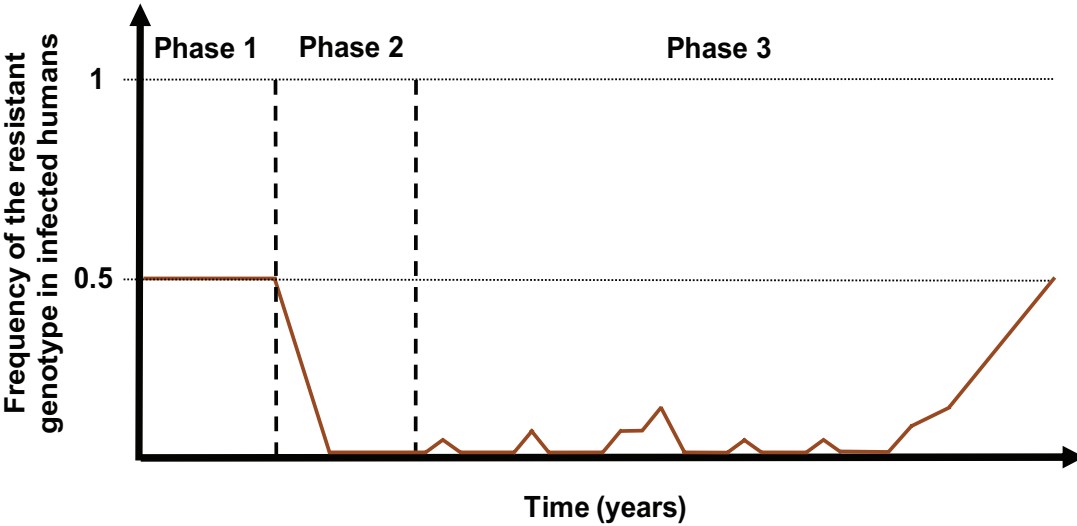

**Appendix 1—figure 13.** Schematic illustration of typical simulations run in OpenMalaria to estimate the probability of establishment of a drug-resistant genotype. The brown curve represents the frequency of the resistant genotypes in inoculations. Phase 1 represents the burn-in phase. In the second phase, we introduced a drug to which resistant parasites were hypersensitive. In the last phase, we imported mutation conferring drug resistance at a low rate until one mutation established.

## 5.1 Calculation of the importation rate for each setting

The importation rate, *I*, (imported infections per 1000 individuals per year), was calculated to mimic a mutation rate of $5 \times 10^{-5}$ mutations per infection per year in each setting as in *Hastings et al., 2020*. This low rate ensured that the newly imported genotype either established or became extinct before a new drug-resistant infection was imported. The importation rate was calculated as:

$$I = \frac{1}{N} 2000 N_i u g,$$

where *N* is the human population size, $N_i$ is the number of infections (i.e. the number of infected people), *u* is the mutation rate per infection (i.e. per transmission), and *g* is the number of malaria generations per year. Thus, $N_i u g$ represents the number of de novo resistant infections imported per year. This number was divided by the human population size and multiplied by 1000 to obtain the number of imported infections per 1000 persons per year. This is multiplied by two, as half of the imported infections were sensitive.

