## [Editor Report]

The authors provide an analysis of how various factors (biological, epidemiological, and treatment) impact the establishment and spread of drug-resistant *Plasmodium falciparum* using a combination of transmission modeling and model emulation. This comprehensive approach to investigating the complex dynamics underlying drug resistance explicitly considers several factors, highlighting their roles in the increasingly important public health question relating to spread of drug-resistant *Plasmodium falciparum*.

---

## [Decision Letter]

**Decision letter after peer review:**

Thank you for submitting your article "The influence of biological, epidemiological, and treatment factors on the establishment and spread of drug-resistant *Plasmodium falciparum*" for consideration by *eLife*. Your article has been reviewed by 2 peer reviewers, and the evaluation has been overseen by a Reviewing Editor and Bavesh Kana as the Senior Editor. The reviewers have opted to remain anonymous.

Essential revisions:

The primary essential revisions for the paper are outlined by the two reviewers, but focus on the clarifying and streamlining results (figures, results presented, naming conventions, etc.) as well as considering some additional aspects of resistance (sequential evolution of resistance, lack of monotherapies actually being used in practice, and recombination). While addressing all of these points as thoroughly as the authors have presented their current results may be outside of the scope of this paper, they should at least be addressed in some manner to help put the research in context and make the overall conclusions more relevant to other researchers and policy makers.

*Reviewer #1 (Recommendations for the authors):*

1. My main suggestion would be to re-think the visual representation of the results. I found Figure 3 to be particularly challenging to parse. I understand the motivation to have the generalized x-axis but it was very difficult for me to understand what was going on and look back and forth with the legend. I suggest thinking through again which results from Figure 3 are most critical to show (are both seasonal and non-seasonal panels needed, when they are so similar?). In addition, some results from Figure 3-supp 1 (low treatment access) were discussed in the text and I felt it would make more sense to promote those to the main text figure. Treatment rate is a very important variable to consider.

2. For Figure 4 (probability of establishment), is it possible to indicate confidence bounds? It seems there is still some stochasticity in the wiggliness of the curves and confidence bounds would help us interpret the differences between curves.

*Reviewer #2 (Recommendations for the authors):*

1. To increase the future impact of your work, consider a triple combination therapy drug profile.

2. I was surprised not to find explicit mention of other complicated agent-based models of malaria in paragraph beginning line 64. Presumably, some of the factors considered jointly in this study have been considered jointly before in these agent-based models. Does the EMOD malaria model have PK/PD components, for example? I would have imagined it does.

3. I wonder if Drug A and B could have more informative names that would aid interpretation of e.g., half-life results? I'm reminded of leaky versus all-or-nothing vaccines in the vaccine modelling literature. Perhaps partner-like and artemisinin-like would suffice.

4. For the drug B mono and combination therapies, did you include Cmax/EC50, adherence and degree of resistance against drug B altogether? Does that not cause a multicollinearity problem in the variance decomposition? It seems more natural to me to include either Cmax/EC50 alone, or adherence and degree of resistance against drug B together. Also, why is Cmax/EC50 referred to as capturing the killing rate (line 246)? This is a unitless ratio. It makes more sense to me to call this a killing effect, as is done in Table 1 and on line 257.

5. In the discussion, consider expanding upon the possible effects of recombination on the observed results (see Public review).

6. Consider adding some more details about the emulator to increase the methodological impact of your work. For example, can you explain why you choose an HGP? Did you try other methods? Can you quantify how efficient it is? You mention 3500 to 11,500 simulations were used to train the emulator; how many simulations would have been required without the emulator? I personally don't understand what the "two random datasets with a sample size of 100,000, with 150,000 bootstrap replicates" are (lines 718-719). Were these datasets used to generate the line plots (Figures 2b and 3 etc.)?

7. Could two genotypes have the same selection coefficient in the same setting but for different reasons? If so, might they have different probabilities of establishment, that would not be identifiable from the selection coefficient alone?

---

## [Author Response]

Reviewer #1 (Recommendations for the authors):1. My main suggestion would be to re-think the visual representation of the results. I found Figure 3 to be particularly challenging to parse. I understand the motivation to have the generalized x-axis but it was very difficult for me to understand what was going on and look back and forth with the legend. I suggest thinking through again which results from Figure 3 are most critical to show (are both seasonal and non-seasonal panels needed, when they are so similar?). In addition, some results from Figure 3-supp 1 (low treatment access) were discussed in the text and I felt it would make more sense to promote those to the main text figure. Treatment rate is a very important variable to consider.

We thank the reviewer for this suggestion which improved the clarity of Figure 3. As highlighted on lines 253-257, we simplified Figure 3 and now show only the most critical drivers of resistance on each panel and remove panels with similar results (seasonal vs. non-seasonal settings and settings with low adherence vs. high adherence to treatment). The previous version of Figure 3 is now provided as a supplementary figure (Figure 3-supplement 1) to allow readers to compare the effect of all factors in all settings (seasonal vs. non-seasonal, low vs. high level of treatment adherence).

Previously in Figure 3 we did not show the results obtained in settings with low access to treatment, as only the fitness cost strongly impacted the rate of spread for all drugs in these settings (these results had previously been shown as a supplementary figure). However, the reviewer’s point was thoughtful, and we have incorporated these results within Figure 3.

2. For Figure 4 (probability of establishment), is it possible to indicate confidence bounds? It seems there is still some stochasticity in the wiggliness of the curves and confidence bounds would help us interpret the differences between curves.

We thank the reviewer for this excellent recommendation. We have included 95% confidence intervals within Figure 4.

Reviewer #2 (Recommendations for the authors):1. To increase the future impact of your work, consider a triple combination therapy drug profile.

We thank the reviewer for this excellent comment and this public review allowed us to clarify the implication of our results concerning triple combination therapy. Our study assessed which factors drive the evolution of resistance to the different drug profiles used for currently implemented ACTs. We showed that resistance to the partner drug (long acting, previously referred to as drug B) depends on the selection window. This result supports the evidence that triple artemisinin combination therapies (TACTs) can delay the spread of resistance to partner drugs as it can minimise the selection pressure that occurs during the selection window if the two long-acting drugs have matching half-lives. While we agree future work could focus on selective pressures from different drug profiles in TACT, this would require a very large study to look at different profiles of three drugs etc. and is outside our scope of already a very large study. Even so, we believe no additional analysis is necessarily needed to highlight the points on TACT above (and in the paper) as they logically follow from our results.

However, we agree that other factors are likely to play a role in the evolution of resistance under TACTs, such as the inverse selection pressure generated by some drugs as highlighted by the reviewer in the public review. Understanding the impact of factors on the evolution of resistance to TACTs is an important question. However, this question is outside the scope of our study, as it would require running many more analyses and considering additional factors (such as the inverse selection pressure generated by some drugs, the synergic effect between drugs, or the fact that some mutation can conferee some degree of resistance to multiple drug, etc.). We agree this should be considered in future work. Nevertheless, it remains worth highlighting that our results imply that TACTs have a great potential to delay resistance to artemisinin and their partner drugs. We edited our discussion to highlight the need for additional analyses to understand which factors may drive the evolution of resistance under TACT (L481-484).

In addition, as pointed out by the reviewer in the public review, monotherapies are no longer recommended. However, we first investigated the driver of resistance to the two drug profiles used in ACT, assuming they were used in monotherapy to identify the determinant specific to each drug profile. Firstly we did this to provide a holistic general understanding of malaria resistance. Secondly this allowed us to identify some determinants that would not have presented themselves had we only examined combination therapy. For each drug profile, once we identified which factors drive resistance, we looked at the drug combinations and observed how the dynamics changed. We have added this information to our manuscript (L134-136).

2. I was surprised not to find explicit mention of other complicated agent-based models of malaria in paragraph beginning line 64. Presumably, some of the factors considered jointly in this study have been considered jointly before in these agent-based models. Does the EMOD malaria model have PK/PD components, for example? I would have imagined it does.

We previously cited a recent study that used three different agent-based models of malaria (MORU, Imperial, and PSU) later in the manuscript. We added this study to the list of models cited in the section highlighted by the reviewer (now L67). However, please note that this study did not systematically assess the joint impact of multiple factors on the establishment and spread of resistance. In addition, EMOD does include PK/PD models, but we did not find a publication that used the EMOD model to investigate the evolution of drug resistance.

3. I wonder if Drug A and B could have more informative names that would aid interpretation of e.g., half-life results? I'm reminded of leaky versus all-or-nothing vaccines in the vaccine modelling literature. Perhaps partner-like and artemisinin-like would suffice.

We thank the reviewer for this helpful recommendation. We have modified the text and figures to replace "drug A" with "short-acting drug" and "drug B" with "long-acting drug".

4. For the drug B mono and combination therapies, did you include Cmax/EC50, adherence and degree of resistance against drug B altogether? Does that not cause a multicollinearity problem in the variance decomposition? It seems more natural to me to include either Cmax/EC50 alone, or adherence and degree of resistance against drug B together. Also, why is Cmax/EC50 referred to as capturing the killing rate (line 246)? This is a unitless ratio. It makes more sense to me to call this a killing effect, as is done in Table 1 and on line 257.

(i) Re the Cmax/IC50 capturing the killing rate sub-question

We initially assessed the effects of Cmax and EC50 of the sensitive genotype independently. We then noted that the impact of the EC50 of the sensitive genotype and the Cmax on the drug killing effect post-treatment depended on their relative values (i.e. the ratio between them) and not their values. When the EC50 of the sensitive genotype is closer to the Cmax, the duration of the drug killing effect on the sensitive genotype is shorter. The drug killing effect also depends on the Emax (which determines the intensity of the killing effect) and drug half-life (which determines the duration of the killing effect). Nevertheless, for fixed Emax and half-life values, the same Cmax/EC50 ratio values lead to the same duration of the drug killing effect irrespective of the values of Cmax and EC50 (see Author response image 1). Thus for simplicity, we reported the impact of Cmax and EC50 as a ratio.

**Author response image 1. sa2fig1:** Illustration of the impact of Cmax/EC50 ratio on the duration of the drug killing effect The figure illustrates the drug killing rate post-treatment of drugs having various Cmax and EC50 values.

We thank the reviewer for highlighting this point. To improve clarity around this point, we added information to the legend of Table 1 and the Results section (L286-298). We also thank the reviewer for pointing out that the term "killing rate" was misused. We replace it with "the duration of the drug killing effect".

(ii) Re the multicollinearity sub-question:

In our analysis, we estimated the Cmax/EC50 ratio based on the EC50 of the sensitive genotype, and thus it did not capture the degree of resistance. Thus note that the intensity and the duration of the drug killing effect for the resistant genotype depend on: Emax, ratio Cmax/EC50, half-life, and degree of resistance (relative increases of the EC50 of the resistant genotype compared to the sensitive one).

The ratio Cmax/EC50 was directly impacted by treatment adherence, as a lower adherence would lead to a lower Cmax. This did not cause a multicollinearity problem because we did not assess the impact of the level of treatment adherence through our global sensitivity analysis (i.e. sample level of treatment adherence from a range and estimate the Sobol indices). However, we ran the global sensitivity for two settings, one with a high level of treatment adherence (patient adhered to 3/3 treatment doses) and one with a lower level of treatment adherence (patient adhered to 2/3 treatment doses). We assessed the effect of treatment adherence by comparing the distribution of selection coefficients across these two settings. Please note that we varied the Cmax in each setting to see how the variation of drug dosage can impact the rate of spread.

5. In the discussion, consider expanding upon the possible effects of recombination on the observed results (see Public review).

The lack of a recombination option in OpenMalaria means we can only investigate genetic variability (drug sensitivity and resistance in our case) at one genetic locus at a time, i.e. we assume there is no genetic variability at other loci. In the case of drug combinations, we track the spread of resistance at the variable locus while assuming that resistance to the other drug is either absent (0%) or is fixed at 100%. We think this is the most likely scenario, i.e. that resistance spreads first to 100% for one drug, then starts to spread to the second. Unfortunately, this lack of recombination is a fundamental technical limitation of OpenMalaria, so we cannot model the simultaneous spread of resistance to two drugs.

Importantly, these assumptions did not impact the probability of establishment and rate of spread of parasites that need only one mutation to confer resistance or do not have a mutation that reduces the fitness cost associated with resistance. However, we now address the points raised by both reviewers by further discussing the main consequences of ignoring recombination in the discussion (L540‐570). In summary, we now state that, in high transmission settings, our model overestimated the probability of establishment of resistant parasites that need multiple mutations to be drug‐resistant or that require additional mutations to restore the fitness cost. This means that the difference between the probability of establishment in high and low transmission settings in such circumstances may be larger than reported here (figure 4). In addition, in high transmission settings, we have probably overestimated the spread of resistant parasites that have multiple mutations involved in the resistant phenotype, especially when these mutations are present in low frequencies, as, under this condition, resistant parasites are more likely to recombine with sensitive parasites than resistant parasites, leading to the separation of these mutations.

6. Consider adding some more details about the emulator to increase the methodological impact of your work. For example, can you explain why you choose an HGP? Did you try other methods? Can you quantify how efficient it is? You mention 3500 to 11,500 simulations were used to train the emulator; how many simulations would have been required without the emulator? I personally don't understand what the "two random datasets with a sample size of 100,000, with 150,000 bootstrap replicates" are (lines 718-719). Were these datasets used to generate the line plots (Figures 2b and 3 etc.)?

We thank the reviewer for highlighting that some of our methodologies needed more detail to increase understanding. We have added the information highlighted below to the manuscript.

We have used HGP as our emulator because it has been used in two previous published studies where global sensitivity analyses were performed using the OpenMalaria model (refs. 97 and 107). In these studies, HGP achieved a good fit between various inputs and outputs used in the model. Reiker and colleagues (2021) (ref. 97) compared the performance of different emulators (results not shown in their paper) to execute global sensitivity analyses using OpenMalaria and found that HGP and neural networks perform the best. However, neural networks require more simulations to reach a satisfactory fit than HGP. Thus, HGP was a better choice for our study to minimise the number of simulations that needed to be run in OpenMalaria. This information was added on lines 811-814.

In our study, HGP accurately captured the relationship between input and output parameters of our model, OpenMalaria. Accuracy was assessed by estimating the root means squared error and the correlation coefficient between selection coefficients predicted with the emulator and selection coefficients obtained from OpenMalaria for a testing dataset (Supplementary file 1, Figures 6-12). This information was explained in the methods section of the original version of our manuscript, but we have moved this section earlier in our manuscript (L814-817) to highlight it sooner and increase understanding.

Without an emulator, each global sensitivity analysis would have required running a very large number of simulations which was computationally infeasible. Here we trained the emulator with 3,500 to 11,500 model simulations. Then we could predict the selection coefficient using the trained emulator and perform the global sensitivity analysis. The emulator outputs were also used to produce Figures 2b and 3. The manuscript text was edited to further clarify this point (L730-736 and L826-L833).

7. Could two genotypes have the same selection coefficient in the same setting but for different reasons? If so, might they have different probabilities of establishment, that would not be identifiable from the selection coefficient alone?

We thank the reviewer for this important question. Two different mutations can have the same selection coefficient for the same setting. In simple situations these two mutations would have the same probability of establishment in this setting as they would be equally selected by natural selection, and the heterogeneity of reproductive success would be the same. However, in some unusual cases, it may not be the case. For example, in our model, one allele could confer high-level resistance but suffer a fitness penalty, while a different allele could confer low-level resistance but without a fitness penalty, and they could end up having the same selective advantage. This could increase transmission heterogeneity in the first allele, which might do very well in drug-treated individuals but very poorly in untreated individuals. We have clarified our explanation of establishment (lines 368 onwards) in response to other Reviewers’ comments which we hope to have identified the general principles.